# Past penguin colony responses to explosive volcanism on the Antarctic Peninsula

Stephen J. Roberts[1,*], Patrick Monien[2,3,*], Louise C. Foster[1,4,*], Julia Loftfield[2], Emma P. Hocking[5], Bernhard Schnetger[2], Emma J. Pearson[4], Steve Juggins[4], Peter Fretwell[1], Louise Ireland[1], Ryszard Ochyra[6], Anna R. Haworth[1,7], Claire S. Allen[1], Steven G. Moreton[8], Sarah J. Davies[9], Hans-Jürgen Brumsack[2], Michael J. Bentley[10] & Dominic A. Hodgson[1,10]

Changes in penguin populations on the Antarctic Peninsula have been linked to several environmental factors, but the potentially devastating impact of volcanic activity has not been considered. Here we use detailed biogeochemical analyses to track past penguin colony change over the last 8,500 years on Ardley Island, home to one of the Antarctic Peninsula's largest breeding populations of gentoo penguins. The first sustained penguin colony was established on Ardley Island c. 6,700 years ago, pre-dating sub-fossil evidence of Peninsula-wide occupation by c. 1,000 years. The colony experienced five population maxima during the Holocene. Overall, we find no consistent relationships with local-regional atmospheric and ocean temperatures or sea-ice conditions, although the colony population maximum, c. 4,000–3,000 years ago, corresponds with regionally elevated temperatures. Instead, at least three of the five phases of penguin colony expansion were abruptly ended by large eruptions from the Deception Island volcano, resulting in near-complete local extinction of the colony, with, on average, 400–800 years required for sustainable recovery.

[1] British Antarctic Survey (BAS), Natural Environmental Research Council (NERC), High Cross, Madingley Road, Cambridge CB3 0ET, UK. [2] Institute for Chemistry and Biology of the Marine Environment (ICBM), Carl-von-Ossietzky-Str. 9–11, 26133 Oldenburg, Germany. [3] Department of Geosciences, Bremen University, Klagenfurter Str. 2–4, 28359 Bremen, Germany. [4] School of Geography, Politics and Sociology, Newcastle University, Newcastle-upon-Tyne NE1 7RU, UK. [5] Department of Geography, Northumbria University, Ellison Building, Newcastle-upon-Tyne NE1 8ST, UK. [6] Institute of Botany, Polish Academy of Sciences, ul. Lubicz 46, 31–512 Kraków, Poland. [7] School of Earth and Ocean Sciences, Cardiff University, Main Building, Park Place, Cardiff CF10 3AT, UK. [8] NERC Radiocarbon Facility (Environment), Scottish Enterprise Technology Park, Rankine Avenue, East Kilbride, Scotland G75 0QF, UK. [9] Department of Geography and Earth Sciences, Aberystwyth University, Aberystwyth SY23 3DB, UK. [10] Department of Geography, Science Laboratories, Durham University, South Road, Durham DH1 3LE, UK. * These authors contributed equally to this work. Correspondence and requests for materials should be addressed to S.J.R. (email: sjro@bas.ac.uk) or to P.M. (email: monien@icbm.de).

Evidence of different penguin species' responses to changes in climate and other environmental variables are based on short-term observational records (typically the last 30–60 years[1–8]), studies of the age and provenance of sub-fossils found in abandoned penguin colonies[9–16], and genetic and genomic studies of their evolutionary history[17–19]. Most of these studies have focussed on how climatic, oceanographic and anthropogenic factors affect penguin populations through access to food and nesting sites and predator–prey dynamics. As far as we are aware, no studies have investigated the long-term impact of large explosive volcanic eruptions on penguin colony size and distribution.

Deception Island off the north-western Antarctic Peninsula (AP) (Fig. 1a) is a highly active volcano[20–24]. Its Late Pleistocene-early Holocene, caldera-forming eruption(s) were exceptionally explosive (Volcanic Explosivity Index (VEI)[25] of 6–7), producing an estimated 30–60 km$^3$ of volcanic ash[24,26], but the 20+ eruptions of the 'historical' era (last c. 200 years) and the 30+ eruptions identified in Holocene (11.75–0 ka) records have been smaller and less explosive. Most small eruptions from Deception Island present an immediate hazard to penguins nesting within its crater and in the immediate vicinity from ash-fall and pyroclastic flows. However, fine volcanic ash is often widely dispersed across the AP by strong Southern Westerly winds[20,22,27,28]. For example, following the December 1967–1970 CE eruptions (c. 0.1 km$^3$, VEI = 3)[20], a fine layer of tephra <2 mm thick was deposited on Ardley Island and the Fildes Peninsula, King George Island and South Shetland Islands (SSI), c. 120 km north-west of Deception Island (Fig. 1a)[21], and in the James Ross Island (JRI) ice core, c. 200 km away on the north-eastern AP[29]. Even relatively minor

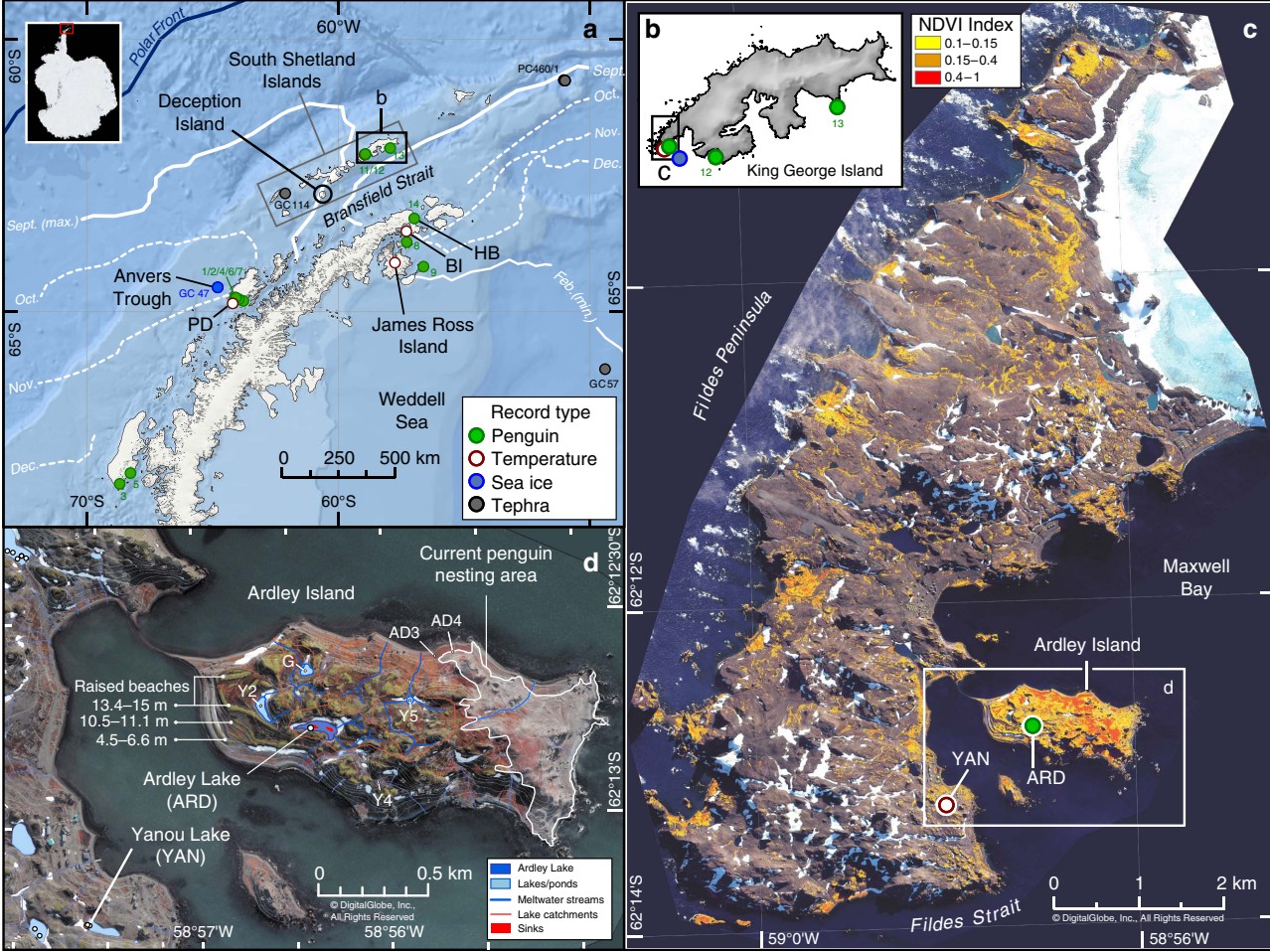

**Figure 1 | Study sites and the location of different types of records examined.** (**a**) Antarctica, the northern Antarctic Peninsula (AP), the Anvers Trough marine core GC047 sampling site and the South Shetland Islands (SSI), located between the mean modern-day (NSIDC 1981–2010 CE; 20% sea-ice coverage) Austral summer minimum (February or Feb.) and winter maximum (September or Sept.) sea-ice limits (solid white lines) in the Bransfield Strait area. The minimum (Feb.) sea-ice extent on the western side of the AP is located off-image further south of the December (Dec.) sea-ice extent (sea-ice data are from https://nsidc.org/data/seaice_index/). This figure illustrates how King George Island becomes sea-ice-free before other parts of the SSI and AP, enabling earlier nearshore sea-ice-edge foraging during the Austral spring/summer; locations 1–13 refer to penguin sub-fossil data shown in Fig. 5c and listed in Supplementary Data 5; other marine core locations used in tephra analysis are shown in Supplementary Fig. 20; BI = Beak Island, HB = Hope Bay, PD = Palmer Deep; (**b**) King George Island (KGI) showing the location of records examined in this study, with legend as in **a**; (**c,d**) Fildes Peninsula, Ardley Island and the study sampling sites at Yanou Lake (YAN, on KGI), Ardley Lake (ARD) and previous study sites (Y2, Y4, G, AD3, AD4) (refs 42–44) on Ardley Island where bio-geochemical evidence of penguin occupation exists, and the location of meltwater pond Y5. The catchment area of Ardley Lake is c. 66,250 m$^2$ and has an elevation range of c. 16–60 m above present sea level (m a.p.s.l.; mean elevation: 31 ± 10 m a.p.s.l.) defined from the King George Island Digital Elevation Model (KGI-DEM). Published raised beach elevations are in m a.p.s.l. as defined in ref. 85. Vegetated areas in **c** were mapped from a Normalized Difference Vegetation Index (NDVI) analysis of satellite data acquired on 4 November 2013. See Supplementary Note 3. This figure includes material copyright of DigitalGlobe, Inc., All Rights Reserved, used with permission under a NERC-BAS educational license and not included in the Creative Commons license for the article.

volcanic eruptions can be potentially devastating to the ecology at locations far from the eruption source[30]. In terms of penguin mortality, the fine basaltic-andesitic tephra, most commonly produced by Deception Island eruptions, contains large volumes of fine and physically abrasive glass, poisonous aerosols (for example, $SO_2$, F), as well as trace elements and toxic metals (for example, Cd, As, Pb)[31] that can adversely affect their physiological functions.

Gentoo penguin (Pygoscelis papua) populations across the AP and elsewhere have remained stable or increased over the last 30 years[5,32,33], while more 'sea-ice dependent' species such as Adélie penguins (Pygoscelis adeliae) have declined[32,34]. These changes have been linked to 'recent' regional atmospheric and oceanographic warming on the AP[7,8]. Conversely, Adélie colony populations are increasing in parts of Antarctica (for example, the Ross Sea) where sea-ice is currently expanding[35], or have increased during previous colder periods[36]. Recent AP-wide reductions in land and sea-ice extent/seasonality have altered access routes to breeding sites and have also caused changes in prey availability by shifting the location and distribution of northern-AP shelf-edge krill spawning grounds[1,3,8,9,35,37]. Since AP krill biomass has not changed significantly over the same period[35,38], recent changes could potentially relate to the early–mid twentieth century 'krill-surplus', which resulted from a reduction in predation pressure on krill due to the exploitation of large numbers of seals and whales in the late nineteenth century to mid twentieth century[16,38,39]. Gentoo penguins prefer to breed in ice-free areas, and are generalist, inshore predators (foraging within c. 20–30 km of their breeding sites), compared to the more krill-dependent, wider-ranging Adélie penguins, while reduced competition from other species on the AP has increased the availability of gentoo prey species across all trophic levels[4,16,40]. Intriguingly, genetic studies have shown that recent AP species-specific population trends are different from those of previous warm-periods, but that the response of gentoo penguins to warming is consistently positive[16,17].

Ardley Island in the SSI (Fig. 1c,d) contains one of the largest breeding gentoo penguin colonies in the Antarctic[41] with c. 5,000 pairs (c. 1.3% of the current global gentoo population), alongside c. 200 breeding pairs of Adélie penguins and <50 breeding pairs of chinstrap penguins (P. antarctica) (Supplementary Fig. 1). Between 1950 and 1997 CE, the north-western AP was one of the most rapidly warming regions in the Southern Hemisphere[7], and between 1980 and 2005 CE the number of breeding pairs of gentoo penguins on Ardley Island increased by c. +310% (minimum–maximum breeding pair count data), whereas between 1980 and 2014, Adélie (−12%) and Chinstrap (−3.7%) penguin populations have both declined (Supplementary Fig. 1). As each penguin produces c. 84.5 g of guano per day[42], every breeding season c. 139 tonnes (dry mass) of penguin guano is discharged onto Ardley Island[42], either accumulating in soils and shallow meltwater ponds and lakes (Fig. 1d), or discharging into the sea. Of these, lakes are protected by a permanent water column and can accumulate sediments without significant disturbance. Geochemical analyses of guano signatures in lake sediments can thereby provide long-term records of past penguin presence in their catchments[43,44] (Supplementary Note 1).

Ardley Lake (62° 12.774 S, 58° 56.398 W; c. 16 metres above present mean sea level, m a.p.s.l., as defined in ref. 45) is the only large (c. 7,270 m²), permanent closed-basin water body on Ardley Island (c. 268,510 m²), and its only permanent depositional 'sink' (Fig. 1d, Supplementary Fig. 2). With two prominent meltwater inflows, and an overspill outflow bounded by its retaining sill at c. 18–16 m a.p.s.l. (max.-min. elevation[45]), the Ardley Lake catchment area (c. 66,250 m²) is well-situated in the centre of

Ardley Island to provide a long-term and continuous record of changes in penguin occupation, because, unlike other shallow ponds on the island (Lake Y1, G, AD3 and AD4, Supplementary Notes 1–3), it has remained above sea-level for the last c. 9 ka (refs 45,46) (Fig. 1d).

Our control site, Yanou Lake (62° 13.243 S, 58° 57.591 W; c. 11 m a.p.s.l.), located opposite Ardley Lake on Fildes Peninsula, is a former shallow marine basin, submerged below sea level until c. 6.5 ka, with basal sediments deposited in a glaciomarine environment between c. 12 and 7.8 ka (ref. 45). The marine embayment evolved into a shallow near-shore lagoonal basin c. 6.5–6.0 ka and, when sea-level fell below the retaining sill at c. 14.5.–11 m a.p.s.l. (max.–min. elevation; ref. 45). Yanou Lake was formed and then developed into a predominantly freshwater lake until the present day. As there is no evidence of past or present penguin occupation in the Yanou Lake catchment, this record enabled us to separate the bio-geochemical inputs associated with penguin guano in the Ardley Lake record from those associated with the underlying geology, soils, tephra deposition and natural lake development.

In this study, we undertook detailed geochemical and multi-proxy analyses on an 8,500-year lake sediment record from Ardley Lake (ARD) and compared the results with a lake sediment record from Yanou Lake (YAN). Using variations in inorganic bio-element geochemistry, we established a novel method to determine the proportion of guano present in the ARD and YAN lake sediments—the ornithogenic sediment fraction or $F_{o.sed.}$. By converting the percentage $F_{o.sed.}$ into guano-influenced sediment dry mass accumulation rates, we modelled estimates of penguin population change for the Ardley Lake colony (see Methods for details). We then determined local-regional drivers of past colony change, by comparing results from the ARD and YAN records with Peninsula-wide records of: (1) lacustrine and sub-fossil evidence of past penguin colony presence[11,43]; (2) palaeoclimate[29], palaeoceanographic[47,48] and relative sea level (RSL)[45] change, including the first AP lake sediment biomarker-based quantitative temperature reconstruction (Yanou Lake), and a new shelf-edge marine sediment record (Anvers Trough) (Fig. 1a; see Methods for details); (3) explosive volcanism from Deception Island, which we conclude had a significant impact on colony population at least three times during the Holocene.

## Results

**Guano geochemistry.** The geochemistry of Ardley Lake (ARD, study site) and Yanou Lake (YAN, control site) sediments showed significant fluctuations with sediment depth and modelled age (Figs 2–4, Supplementary Notes 4–7, Supplementary Figs 3–13, 16, Supplementary Tables 1–7, Supplementary Data 1). R-mode cluster analysis performed on the inter-correlation coefficients of major and trace element concentrations in the ARD record clearly separated guano-derived elements (Sr, Cu, Zn, Se, Ca, Se, P, C, N, S) and guano-associated elements (Cd, As, Hg) from lithogenic elements (for example, Al, Mg, Si, Sc, Ti, Zr, Y and REE) derived from weathering of local volcanic bedrock (Fig. 2, Supplementary Fig. 8b, Supplementary Table 4). No similarly clear separation of elements was evident in the YAN control record (Supplementary Fig. 8c; Supplementary Table 5). The positive correlation between Ca and P ($r = 0.63$, $P < 0.001$; Supplementary Table 4) suggests that hydroxylapatite is the main phosphate phase in the ARD record (Supplementary Notes 1,8). Selected element-aluminium cross-plots (Fig. 3a) show that Ardley Lake sediments clearly represent a mixture between two end members: eroded Ardley Island bedrock and ornithogenically derived (bird-formed) soil and sediment from the lake's catchment, whereas the geochemistry of Yanou Lake sediments is

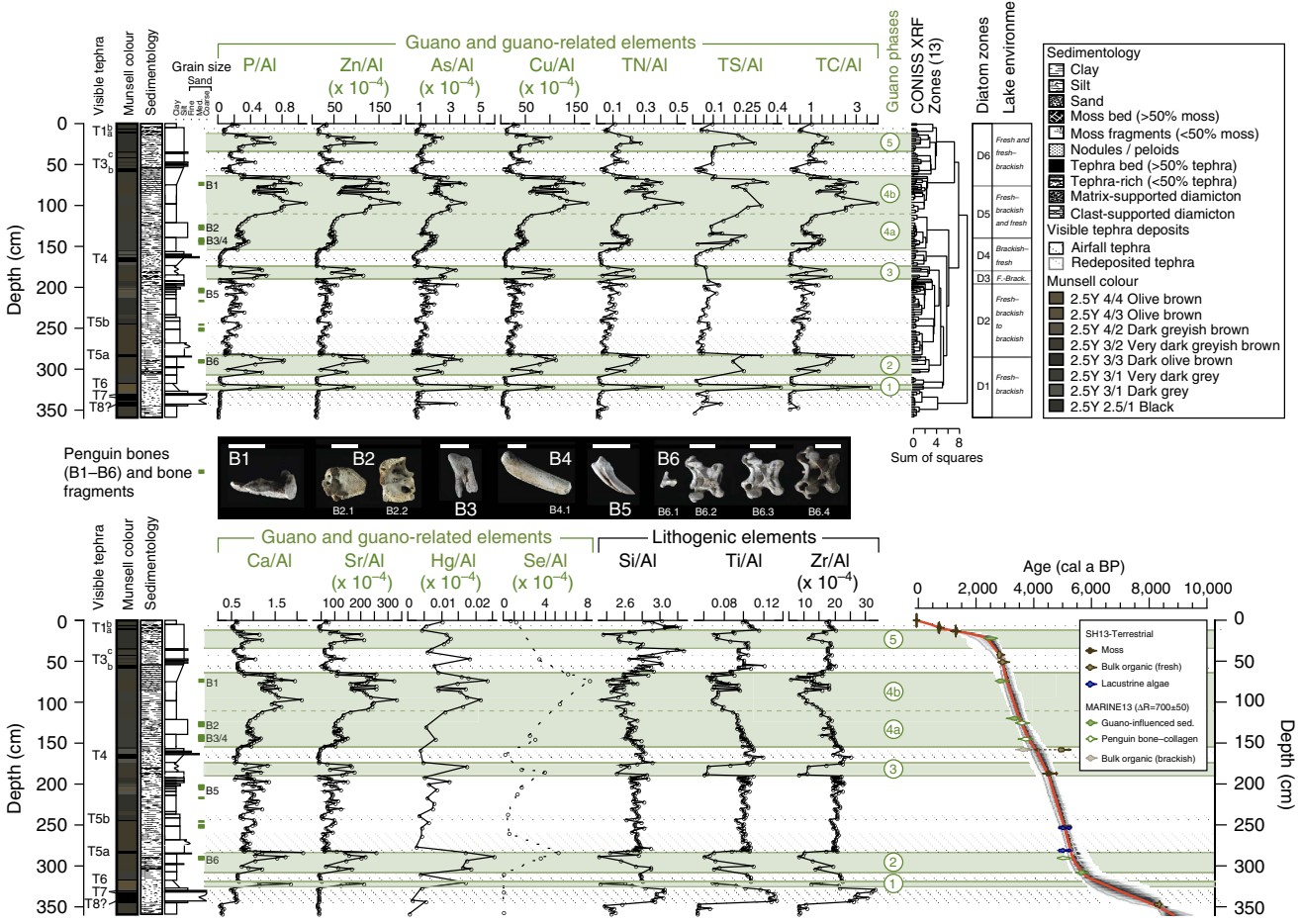

**Figure 2 | Whole core geochemistry of Ardley Lake sediments.** Downcore profiles of selected aluminium normalized guano-elements (P, Zn, As, Cu, TN, TS, TC, Ca, Sr, Hg, Se), lithogenic elements (Si, Ti, Zr) and age-depth model (ARD-M5) of the ARD record (Supplementary Data 1,2). Intact juvenile *Pygoscelis Papua* bones found at core depths are labelled B1–B6 (scale bars are 1 cm): B1 is a cnemial crest bone of the tibiotarsus (left leg); at B2, two base occipital condyle bones from a poorly fused skull(s); B3 is a heavily weathered metatarsal (foot) bone; at B4, two weathered carpometacarpus (wing) bones; B5 is a talon; B6 are mid-upper neck cervical vertebrae bones, found intact vertically (4 of 5 bones shown). Bone-collagen was extracted and radiocarbon dated from bones B2.1, B4.1, B6.2 (see Supplementary Fig. 4 and Supplementary Table 2 for details).

strongly associated with local bedrock and tephra deposition. The influence of penguin guano on the sediment samples from Ardley Lake is further supported by their C/N and C/P ratios (C/N = av. 6.6, C/P = av. 3.3), which are close to those previously reported in ornithogenic soils (C/N = 6.3, C/P = 1.2–3.7) (ref. 49) (Fig. 3b). In contrast, they differ significantly from ratios found in phytoplankton and local plants, for example, carpet mosses, lichens (*Usnea antarctica*), liverworts and vascular plants, such as *Deschampsia antarctica* (Fig. 3b, C/N = 21–114)[50] present in the catchment area of both lakes.

In the ARD record, in-phase changes in several aluminium-normalized bio (associated) elements, such as P, Zn, As, Cu, Ca, Sr, Hg, and Se, together with TN, TC (equivalent to TOC) and TS were recorded (Fig. 2, Table 1, Supplementary Figs 6–8,11b,16). This enabled us to derive the mean relative proportion of guano or ornithogenically derived sediment (% $F_{o.sed.}$) in the Ardley Lake sediment matrix using a mixing equation (equation (1, Methods) based on the element/aluminium (Al) ratio of selected bio-elements (Cu, P, Sr, Zn) in the Ardley Lake sediments, the mean Al-normalized Ardley Island bedrock[51] composition and Al-normalized mean composition of guano-bearing ornithogenic sediment and soils[49] from King George Island as end members (see Methods for details).

**Guano phases.** Sustained high concentrations of bio-elements (weighted mean $F_{o.sed.} > 10\%$) and stratigraphically constrained incremental sum of squares (CONISS) cluster analysis were used to define five phases of elevated guano flux (GP-1 to GP-5) (shown as 1–5 in circles in Figs 2, 4 and 5; see Methods for details). Periods of exceptionally elevated guano within each guano phase (darker green shading in Fig. 4b) were defined as > the upper bound (95% confidence level) of the whole-record weighted mean $F_{o.sed.}$ value of 27.93% (green dotted line in Fig. 4b).

The first occupation of the Ardley Lake catchment by penguins occurred between 6.7 and 6.3 cal ka BP (GP-1), with further $F_{o.sed.}$ maxima between 5.8–5.3 (GP-2), 4.5–4.3 (GP-3), 4.0–3.0 (GP-4) and 2.7–1.3 (GP-5) cal ka BP (Figs 4b and 5, Table 1). No similar patterns were found in the control site YAN (Fig. 4a, Supplementary Figs 10,11c). Intact juvenile gentoo penguin bones and bone fragments were found in guano phases GP-2 and GP-4 and in sediments deposited between GP-2 and GP-3 (Fig. 2, Supplementary Fig. 4). Guano dry mass accumulation rates ($F_{o.sed.}$ DMAR in g cm$^{-2}$ a$^{-1}$, thick black line in Fig. 4b) account for variations in the sedimentation rate that reflect changes in erosional input into Ardley Lake, and, therefore, do not always correspond to higher guano-influenced sediment $F_{o.sed.}$ percentages. Since $F_{o.sed.}$ DMAR values underpin

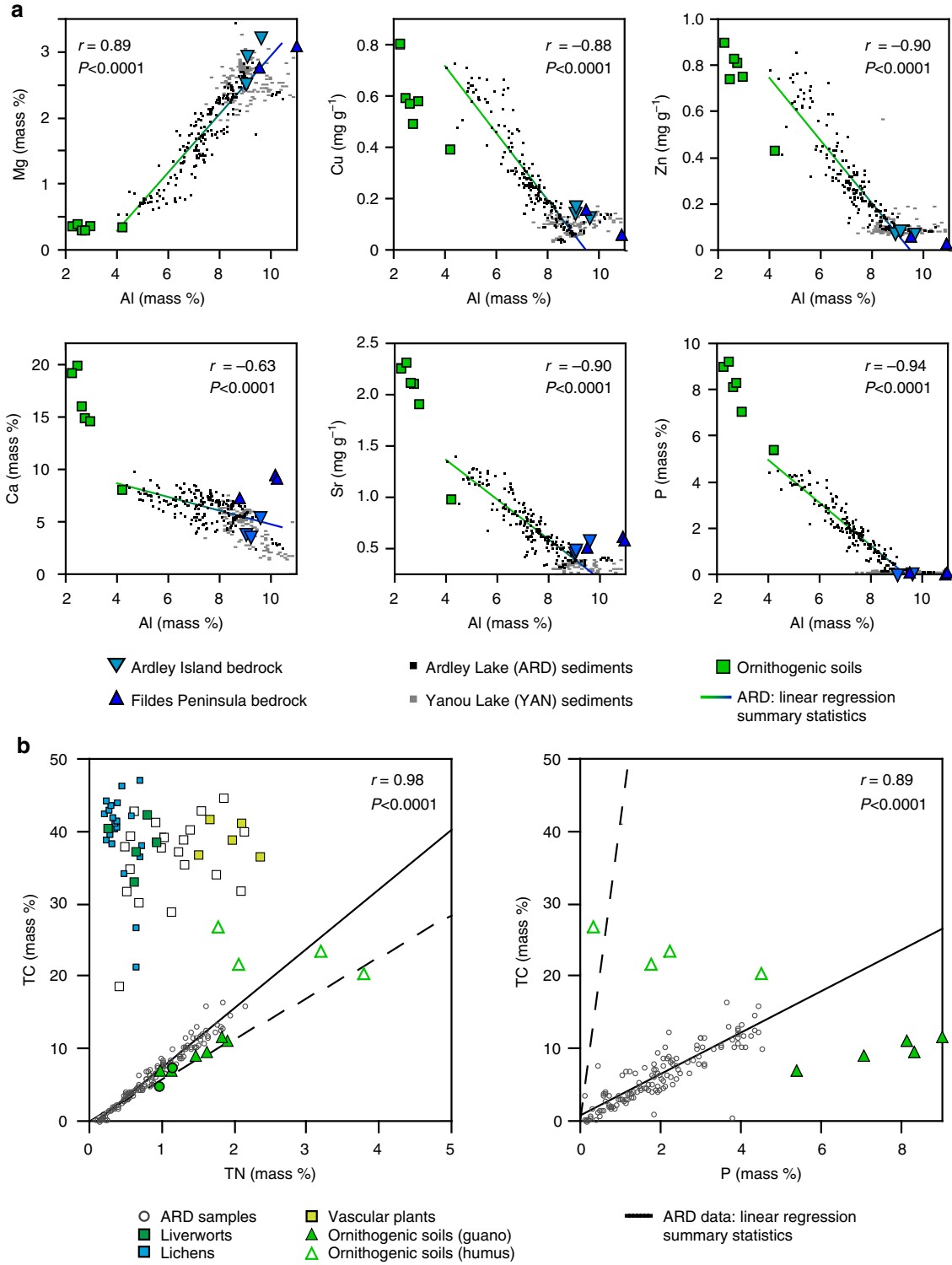

**Figure 3 | Scatterplots of element/aluminium ratios and comparison of C/N and C/P ratios.** (**a**) Scatterplots of selected element/aluminium ratios of Ardley Lake and Yanou Lake sediments (this study), ornithogenic soils[49] and local bedrock[86]. (**b**) Total Carbon/Total Nitrogen (TC/TN) and Total Carbon/Phosphorus (TC/P) ratios of Ardley Lake sediments, local lichens (for example, *Usnea antarctica*)[50], liverworts (*Cephalozia varians*), mosses[50], vascular plants (*Deschampsia antarctica*)[50] and different types of ornithogenic soils[49]. The solid and the dashed lines show linear regression lines of samples from core ARD, and C/N and C/P ratios of phytoplankton according to the Redfield ratio, respectively (Supplementary Data 3).

population models, guano phases GP-2, GP-3 and GP-4 correspond to a higher number of penguins present in the Ardley Lake catchment shown in Fig. 5b (see Methods, Supplementary Note 8, Supplementary Figs 16–19, Supplementary Table 9 for more

details). A good correspondence between the Ardley Lake guano/ fossil record and the sub-fossil record of former penguin colonies from across the AP exists during these phases, and until the end of guano phase GP-5, c. 1.3 cal ka BP (Fig. 5).

**Palaeoenvironmental records**. We then compared these five phases of elevated guano flux with a range of local and regional records of past environments to identify the major drivers of changes in penguin populations. First, we found no significant difference (5 or 10% level) in temperature data during guano and non-guano phases from cross-northern Peninsula 9–0 ka terrestrial temperature records, YAN-GDGT reconstructed mean summer air temperatures (MSAT) (Fig. 4a)

and JRI atmospheric temperature (Fig. 4f)[29]. Overall, none of the guano phases were statistically related to warmer sea surface temperatures in the Palmer Deep (PD) record. Conversely, the non-guano phases were associated with significantly warmer sea surface temperatures in the Palmer Deep record (PD-SST 0–200 m; Fig. 4e)[48] (two-tailed Mann–Whitney $U$-tests: YAN-GDGT: $P = 0.58$, JRI: $P = 0.57$, PD-SST: $P = 0.01$; Supplementary Discussion, Supplementary

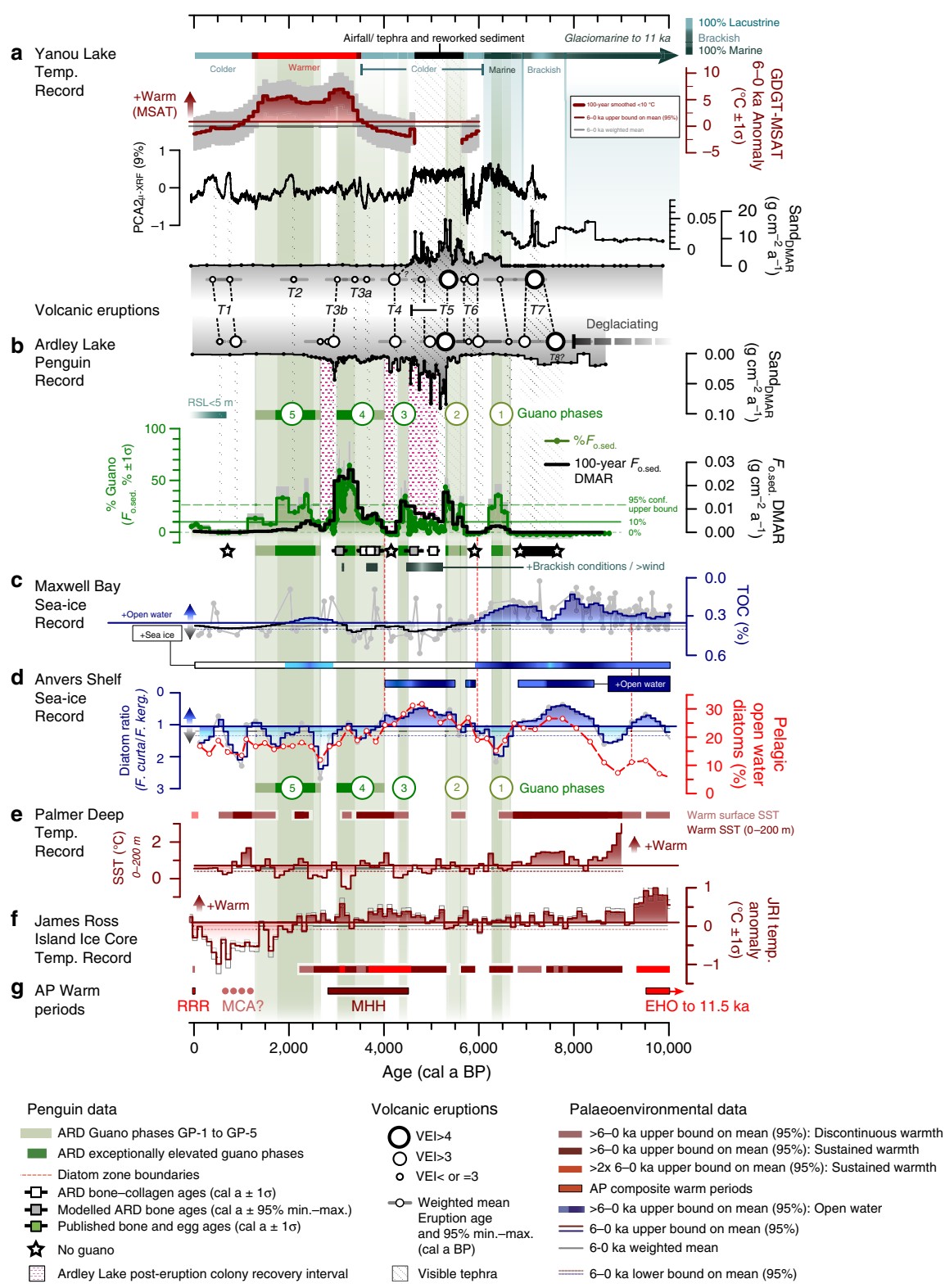

Table 8). Notably, the late Holocene guano $F_{o.sed.}$ DMAR maxima in guano phase GP-4 ($F_{o.sed.}$ % = 43.66 ± 15.37 (1σ) %), occurred between 3.4 and 3.0 cal ka BP when both the YAN-GDGT and JRI records were significantly warmer than their 6–0 ka means (Fig. 4, Supplementary Fig. 18, Supplementary Tables 7,8).

Second, we compared the guano phases with sea-ice and open water proxies, which impact on the penguins' access to nesting sites and prey species. Although guano-phase GP-4 coincided with a sustained period of reduced spring/summer open water seasonality in the Anvers Trough record from c. 4 cal ka BP onwards (Fig. 4d), there were no significant differences in open water conditions in Maxwell Bay ($P = 0.26$) or at the Anvers Shelf ($P = 0.25$) for guano and non-guano phases. More prolonged spring/summer open water conditions existed at the Anvers Trough shelf-edge in mid-Holocene guano phases (GP-2 and GP-3), but not during late Holocene guano phases (GP-4 and GP-5) ($P = 0.10$).

**Volcanic activity.** Third, we compared the guano phases with the combined record of volcanic tephra in the ARD and YAN lake sediment records. We identified seven phases of VEI = 3 or VEI > 3 eruption activity (T1–T7) (Fig. 4a,b). Eruption T7 aside,

mid-late Holocene tephra layers had well-defined basaltic-andesitic compositions, typical of Deception Island source compositions (Supplementary Figs 20,21, Supplementary Table 10). Three VEI > 3 eruption events (T5a, T4, T3b) coincided with immediate and significant reductions in guano deposition (that is, $F_{o.sed.}$ to < 10%) (Fig. 4a,b, Table 1). The most disruptive series of eruptions (T5a,b) recorded in both ARD and YAN sediments occurred during the mid-Holocene between c. 5.5–4.9 cal ka BP (combined ARD–YAN 95% maximum to mean age range; Fig. 4, Table 2, Supplementary Figs 12,16, Supplementary Table 3). Brackish conditions, greater catchment destabilization and increased erosion occurred for c. 1,000 years during the T5 eruption series (Fig. 4, Supplementary Fig. 16). The T4 (c. 4.5–4.2 cal ka BP) and T3b (c. 3.2–3.0 cal ka BP) eruptions that followed the T5 eruption series were smaller, but also coincided with immediate and significant reductions in guano deposition. Colony recovery after the T5a eruption took 780 ± 50 years [530–1,130 years], and 410 ± 30 [0–760] years and 480 ± 60 [160–390] years after the T4 and T3b eruptions, respectively, with an overall weighted mean ( ± s.e.) recovery time of 570 ± 110 years (Table 2).

Two other VEI = >3 eruptions (T3a, T2) occurred at the same time as measurable, but not-significant reductions in the

**Table 1 | Summary statistics of guano phases GP-1 to GP-5 timing and duration and guano-influenced sediment percentage data.**

| ARD CONISS zone | ARD Guano phase | ARD Core depth min.–max. (cm) | Calibrated ages (cal a BP) | | Duration (years ± 95% conf. diff.) [95% min.–max.] | n | % Guano-influenced sediment | |
| | | | Weighted mean age min.–max. | 95% conf. age range min.–max. | | | Weighted mean $F_{o.sed.}$ (% ±1σ) | 95% Conf. $F_{o.sed.}$ Interval (lower–upper %) |
|---|---|---|---|---|---|---|---|---|
| 12 | 5 | 13–35 | 1,290–2,670 | 1,160–2,830 | 1,380 ± 145 [1,670] | 11 | 23.57 ± 10.74 | 16.35–30.79 |
| 9 | 4b | 65–112 | 3,000–3,430 | 2,880–3,580 | 1,010 ± 170 [1,350] | 26 | 43.66 ± 15.37 | 39.47–47.85 |
| 10 | 4a | 112–155 | 3,430–4,010 | 3,270–4,230 | | 26 | 12.95 ± 2.76 | 11.41–14.49 |
| 7 | 3 | 175–190 | 4,310–4,510 | 4,090–4,640 | 200 ± 175 [550] | 7 | 26.33 ± 12.16 | 18.92–33.74 |
| 5 | 2 | 284–310 | 5,300–5,750 | 5,180–6,020 | 450 ± 195 [840] | 9 | 29.38 ± 9.90 | 21.95–36.82 |
| 3 | 1 | 320–326 | 6,270–6,670 | 5,820–7,360 | 400 ± 570 [1,540] | 3 | 26.77 ± 9.49 | 5.31–48.23 |
| Summary statistics | | | | Whole record | — | 198 | 25.31 ± 9.41 | 22.69–27.93 |
| | | | | Guano phases | 690 ± 250 [1,190] | 82 | 34.19 ± 12.19 | 31.06–37.32 |
| | | | | Non-guano phases | — | 116 | 3.56 ± 2.49 | 2.42–4.69 |
| | | | | Late Holocene guano phases | 1,200 ± 160 [1,510] | 62 | 36.57 ± 12.73 | 32.73–40.40 |
| | | | | Mid Holocene guano phases | 350 ± 310 [980] | 20 | 27.77 ± 10.72 | 23.66–31.88 |

All ages are in calibrated (equivalent to calendar) years before present (cal a BP), where BP is defined as 1950 CE (see Supplementary Methods); [95% min. max.] is the minimum and maximum 95% confidence age range based on 2.5% and 97.5% quantiles from Bayesian age-depth modelling (see Supplementary Notes 5–7 and Supplementary Methods); durations are in calendar years with guano phase 4 shown as the combined duration of 4a and 4b; 95% conf. diff. (cal years) is the average of the weighted mean minimum −95% confidence minimum ages and the 95% confidence maximum – weighted mean maximum ages; the duration of the 95% confidence minimum to maximum age range is shown in square brackets. Guano-influenced sediment ($F_{o.sed.}$) weighted mean percentages ±1σ, significance level α = 0.05..

**Figure 4 | Penguin occupation phases in the Ardley Lake record compared with key records of volcanic activity climate and sea-ice from the Antarctic Peninsula region for the last 10,000 years.** In order from top: (**a**) Yanou Lake (YAN) sediment core palaeoenvironmental summary, reconstructed mean summer air temperature (MSAT) GDGT 100-year temperature anomaly data; 200 μm micro-XRF (μ-XRF) Principal Component Analysis 2nd axis (PCA2, 9% variance explained) and sand Dry Mass Accumulation Rate (DMAR); main visible tephra deposits are black diagonally hatched zones, dotted lines show correlation of tephra layers between the ARD and YAN records, circle size reflects Volcanic Explosivity Index (VEI) of major eruptions and circle position is the weighted mean modelled age with 95% min.–max. confidence age ranges (grey horizontal bars) based on 2.5% and 97.5% quantiles from Bayesian age-depth modeling (see Supplementary Notes 5–7 and Supplementary Methods). (**b**) Numbered, green-shaded guano phases GP-1 to GP-5 with the calculated fractions of ornithogenic sediments in the Ardley Lake record ($F_{o.sed.}$ % ±1σ errors in grey) overlain by $F_{o.sed.}$ dry mass accumulation rate (DMAR) (thick black line, 1σ errors not shown for clarity, but in GP-4 and GP-5 these are on average c. 25–35%); red stippled areas are recovery intervals; white squares are radiocarbon-dated penguin bone-collagen extract ages; grey squares are modelled ages of bones in the ARD record. (**c**) Maxwell Bay (MB) percentage Total Organic Carbon (TOC) data (in grey) and 100-year interval negative exponential smoothed data (dark blue) used as sea-ice proxy[52]; data > 6–0 ka upper bound on the mean (95% confidence level) represents increased open water conditions (shaded dark blue). (**d**) Anvers Trough GC047/TC046 sediment core: the commonly used diatom-based ratio (*Fragilariopsis curta*: *Fragilariopsis kerguelensis*) with data 6–0 ka upper bound on the mean (95% confidence level) shaded dark blue representing increased open-water conditions and percentage of total spring/summer pelagic open-water diatoms species (red line). (**e**) Palmer Deep (PD) Sea Surface Temperature (SST) 100-year interval records integrated over 0–200 m depth[48] and at the surface[47] with data > 6-0 ka upper bound on the mean (95% confidence level) shaded in dark red; (**f**) James Ross Island (JRI) ice core temperature anomaly record (compared to 1961–1990 CE interval) with s.e. ranges of the isotope-temperature dependence shown as grey lines[29]. (**g**) AP regional palaeoclimate synthesis[56] (Supplementary Data 4).

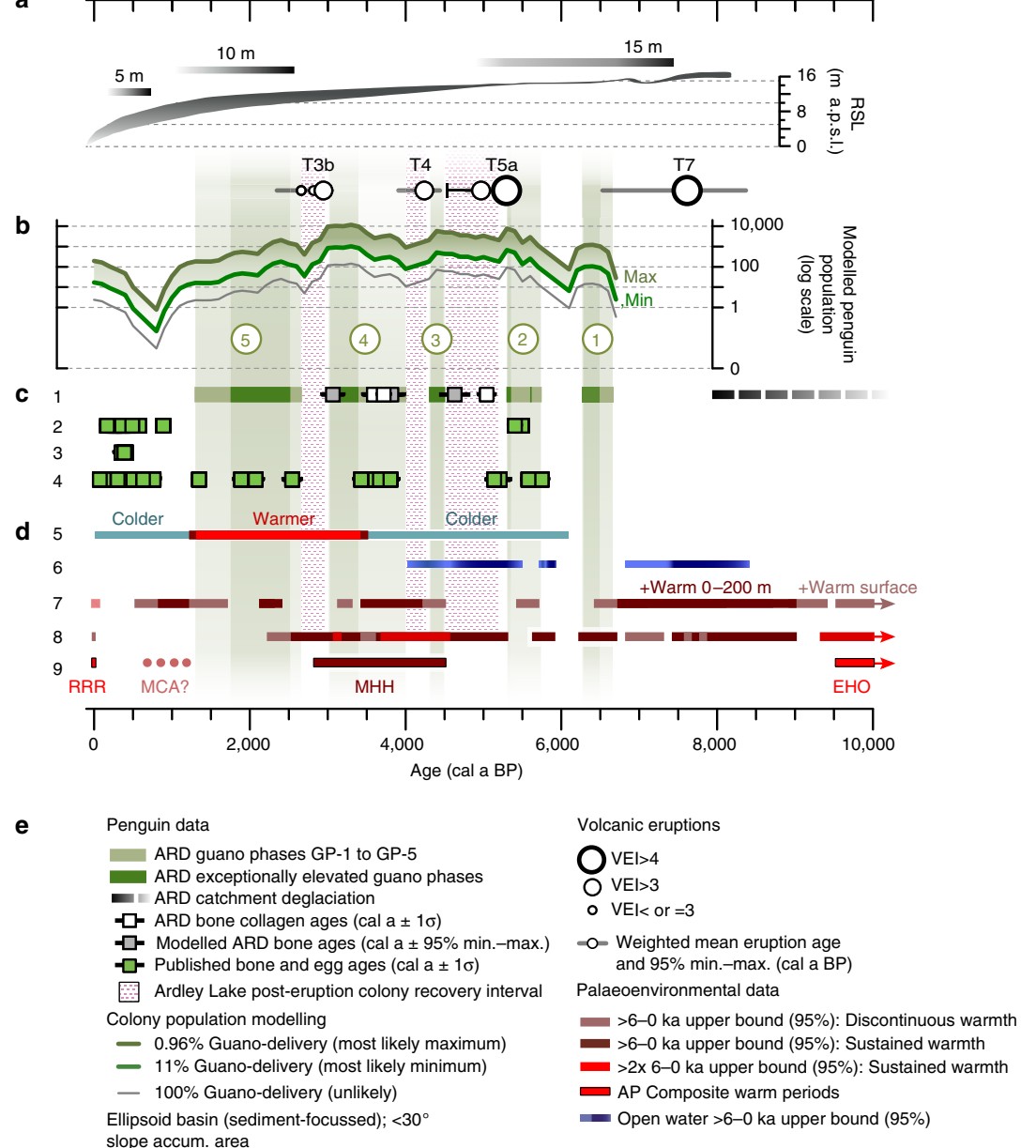

**Figure 5 | Modelled penguin population changes for the Ardley Island penguin colony compared with a summary of key environmental influences during the Holocene.** (**a**) Holocene relative sea level (RSL) changes for the South Shetland Islands (SSI)[45] in metres above present mean sea level (m a.p.s.l.). (**b**) Modelled Ardley Lake (16 m a.p.s.l.) penguin colony population changes in guano phases GP-1 to GP-5 (this study) with major colony impact eruptions (circles), and post-eruption recovery intervals (red stipple); estimated modelled population combined errors for the 0.96% and 11% guano-delivery models are not shown for clarity, but typically 30–50% (Supplementary Fig. 19). (**c**) Sub-fossil penguin occupation record from the Antarctic Peninsula (AP) based on radiocarbon dates of remains at abandoned nesting sites (refs 10,12,53): 1 = Ardley Lake colony (this study), 2 = SSI and the Northwest AP, 3 = Northeast AP, 4 = Mid-southern AP (Supplementary Data 5). (**d**) Summarized timing of warmer phases and periods of greater open water around the SSI and the AP[48,29,56] over the last 10,000 years, defined as > 6–0 ka upper bound on their respective means (95% confidence level), where: 5 = Yanou Lake mean summer air temperature (MSAT) (this study), 6 = Anvers Trough open water record (this study), 7 = Palmer Deep sea surface temperature (SST)[48], 8 = James Ross Island temperature anomaly[29], 9 = AP warm periods[56] (Supplementary Data 4). (**e**) Legend.

percentage of guano in the sediments, while the T6 eruption c. 6.0–5.9 cal ka BP [6.4–5.7 cal ka BP max.–min. 95% confidence interval age range] occurred c. 200 years after the guano phase GP-1 decline had begun (Fig. 4a,b, Supplementary Note 9, Supplementary Discussion, Supplementary Table 3). Eruptions in the early Holocene (T7) and in the last c. 2,000 years (T1,T2) did not have a measurable impact on the Ardley Lake penguin colony as the populations were already at, or near, minima during these times.

## Discussion

Ardley Lake began accumulating sediments c. 8.8 cal ka BP [9.2–8.4 cal ka BP] following retreat of the SSI Ice Cap from this part of the Fildes Peninsula between 10.1 and 8.2 cal ka BP (ref. 52), driven by early Holocene thermal maximum conditions recorded in regional marine and ice core temperature records (the early Holocene Optimum (EHO) in Fig. 4g (refs 29,47,48)). The highly explosive, T7 eruption, c. 7 cal ka BP [6.0–7.7 cal ka BP], had a bi-modal basaltic-andesitic and trachydacitic-rhyolitic

**Table 2 | Major mid-late Holocene eruptions of the Deception Island volcano and Ardley Lake penguin colony recovery interval summary statistics.**

| Volcanic eruption data | | | Guano phase data | | | | |
|---|---|---|---|---|---|---|---|
| Tephra layer | Eruption size | ARD & YAN Age 95% max.–mean (cal a BP) | Decline to $F_{o.sed.}$ <10% | | Recovery guano phase | Recovery to $F_{o.sed.}$ >10% | Weighted mean |
| | | | Guano phase ended | Weighted mean age [95% max.–min.] (cal a BP) | | Weighted mean age [95% max.–min.] (cal a BP) | Recovery interval (years ± 95% conf. diff.) [min.–max.] |
| T5a | VEI>4 | 5,530–5,360 | 2 | 5,280 [5,430–5,160] | 3 | 4,500 [4,630–4,300] | 780 ± 50 [530–1,130] |
| T4 | VEI>3 | 4,480–4,230 | 3 | 4,310 [4,500–4,090] | 4 | 3,900 [4,090–3,740] | 410 ± 30 [0–760] |
| T3b | VEI>3 | 3,220–2,990 | 4 | 2,980 [3,090–2,860] | 5 | 2,490 [2,700–2,160] | 480 ± 60 [160–390] |

Only eruptions that had a significant impact on colony population levels are shown, where significant is defined as a reduction in the proportion of guano-influenced sediment (% $F_{o.sed.}$) to <10% within 200 years after an eruption event; the combined ARD and YAN 95% max. mean age was used to determine the most likely eruption age and is the maximum 95% confidence age weighted mean age, derived from Bayesian age-depth modelling (see Supplementary Notes 5–7, Supplementary Methods); cal a BP is calibrated (or calendar) years before present, where BP is defined as 1950 CE (Supplementary Methods); [95% max.- min.] is the maximum and minimum 95% confidence age range based on 2.5% and 97.5% quantiles from Bayesian age-depth modelling; weighted mean recovery interval duration (cal years) = weighted mean decline age - weighted mean recovery age; 95% conf. diff. is the average difference between the weighted mean recovery interval and its respective 95% max. or min. decline max. or min. recovery interval (cal years); [min.- max.] is the absolute minimum to maximum recovery interval duration (cal years), where the minimum recovery interval duration = 95% min. decline - 95% max. recovery age and the maximum recovery interval duration = 95% max. decline −95% min. recovery age; the 95% conf. diff. weighted average recovery interval duration ( ± s.e.) = 570 ± 110 cal years (1σ ± 160 cal years; 95% lower-upper confidence interval = 90–1050 cal years; significance level α = 0.05; age data are rounded to the nearest 10 years, but rounding to the nearest 100 years provides a more realistic assessment of age-depth modelling errors and comparison to 100-year interval data in Figs 4 and 5. See Supplementary Table 3 for ARD and YAN tephra deposits age data.

composition, and dispersed tephra widely across the northern Peninsula and Scotia Sea (Supplementary Figs 20,21). This and other more widespread early Holocene eruptions occurred before the Ardley Lake colony was established, and could have disrupted early post-deglaciation penguin colonization across the north-western AP (Supplementary Discussion).

Following the EHO, an extended period of generally 'warmer' interglacial conditions, with minor variations in temperature existed on the AP between 9.2 and 2.5 cal ka BP (refs 48,29) (Fig. 4f,g). These conditions, together with the extended periods of open water in Maxwell Bay (Fig. 4c) and on the Anvers Shelf (Fig. 4d) enabled the establishment of a small penguin colony on Ardley Island from c. 6.7 cal ka BP [7.4–5.8 cal ka BP]. This is the oldest *in situ* evidence of a Holocene penguin colony on the AP, and up to c. 1,000 years before the sub-fossil evidence of occupation for the mid-southern western AP from c. 5.8 cal ka BP [5.9–5.6 cal ka BP], and the northern AP from c. 5.5 cal ka BP [5.6–5.3 cal ka BP] (refs 12,53) (Fig. 5).

The peak in penguin populations during guano phase GP-1 (6.7–6.3 cal ka BP) coincided with a continuation of the local open water conditions in Maxwell Bay (Fig. 4c), but at the same time there were contrasting reductions in shelf-edge spring/summer open water conditions 6.7–6.0 cal ka BP in the Anvers Trough (inferred from higher *Fragilariopsis curta/F. kerguelensis* diatom ratios and lower percentage of pelagic open water diatoms), and other records of 'cooler' and increased 'sea-ice' conditions around the SSI 7.3–5.2 cal ka BP (ref. 54). Local sea-ice conditions appear to be the most significant factor in driving GP-1, which ended when there was a gradual increase in sea-ice in Maxwell Bay more than 200 years before the T6 eruption event.

The following two (of three) prominent Ardley Lake guano dry mass accumulation maxima occurred during GP-2 and GP-3 c. 5.8–5.3 and 4.5–4.3 cal ka BP (Fig. 4b, Supplementary Table 7). In marked contrast to GP-1, both GP-2 and GP3 occurred during a period of greater sea-ice extent or seasonality in Maxwell Bay (Fig. 4c), despite the preference of gentoo penguins for ice-free conditions. These maxima also show no strong or consistent relationships with sea-ice conditions on the Anvers Shelf (Fig. 4d), ocean temperatures at PD (Fig. 4e) or atmospheric temperatures in the JRI ice core (Fig. 4f), although the peak of GP-3 corresponds with the onset of a 'mid Holocene Hypsithermal' (MHH in Fig. 4g) at c. 4.5 ka in several terrestrial cross-Peninsula records[55,56]. Instead, the main driver of these penguin population changes is volcanism, with colony populations increasing during volcanically inactive periods and then experiencing abrupt catastrophic declines following major eruptions (Fig. 4b).

After the T5a Deception Island eruption abruptly ended guano phase GP-2, the Ardley Lake colony struggled to fully re-establish itself for c. 800 years (Table 2) due to a series of closely spaced eruption events (Fig. 4b) and a phase of continued sea-ice presence in Maxwell Bay (Fig. 4c). Our analysis of longer-term (100-year interval) trends in the Maxwell Bay sea-ice reconstruction[52] (Fig. 4c) show that these cooler oceanographic conditions persisted well into the late Holocene (5.9–2.6 ka), corresponding to 'cooler' conditions and increased sea-ice in the Bransfield Strait between c. 5 and 2 cal ka BP (ref. 57), and advancing glacier margins, increased local snow/ice and sea-ice and limited primary production in surface waters of King George Island[54]. Despite these cooler conditions the penguin population was able to re-establish itself during GP-3, but was then abruptly terminated by the T4 Deception Island eruption at c. 4.5–4.2 cal ka BP.

The most sustained guano phase, GP-4 (c. 4.0–3.0 cal ka BP) (Table 1), occurred during the warm conditions of the late MHH (Fig. 4g), and is characterized by a marked increase in MSAT reconstructed in the Yanou Lake GDGT record (Fig. 4a), positive temperature anomalies at JRI (Fig. 4f) and a sustained period of significantly warmer cross-Peninsula terrestrial temperatures from c. 3.8 cal ka BP (refs 29,48) (Fig. 4a,f,g, Supplementary Tables 7,8). Warmer conditions have also been recorded at this time in lake and moss peat bank records from Livingston Island, Elephant Island, Beak Island[56,58], some marine records from the AP[48,56,59], Antarctic ice core composite records[60] and stacked Southern Hemisphere temperature records[61] (Fig. 4e–g, Supplementary Fig. 17e–i). These 'more favourable' climatic conditions for rearing juvenile penguins likely drove the Ardley Lake penguin colony to its Holocene population maxima, and led to the re-establishment of colonies in the mid-southern Peninsula region (Fig. 5c). Despite the gentoo penguins' preference to feed near their colony, it appears that GP-4 was not adversely affected by the persistence of sea-ice in Maxwell Bay (Fig. 4c) and around the Anvers Shelf (Fig. 4d) and the cooler sea surface anomalies at PD (Fig. 4e). However, as with guano phases GP-2 and GP-3, guano phase GP-4 was abruptly terminated by a Deception Island eruption event (T3b).

We link the failure to re-establish another large and sustained colony of similar size and duration as guano phase GP-4 to the start of a marked 'neoglacial' decline in temperature at c. 2.5 cal ka

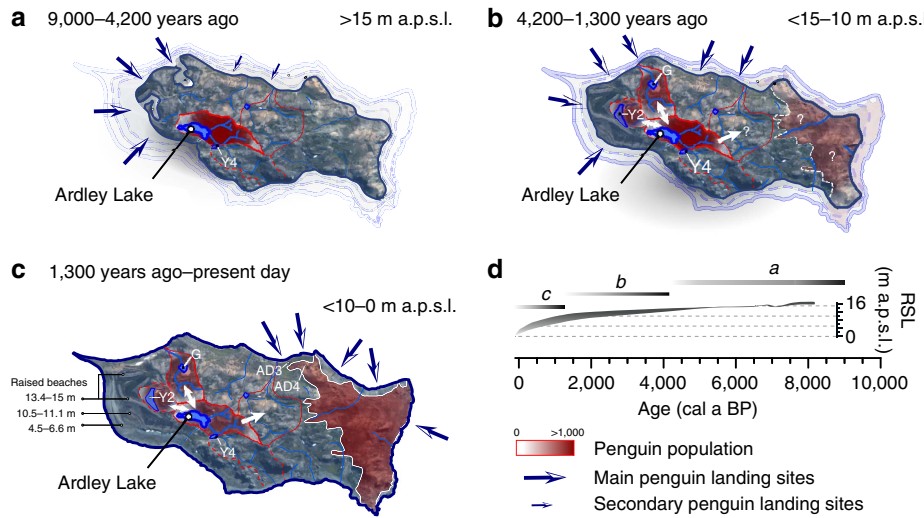

**Figure 6 | Changes in land availability and colony population on Ardley Island over the last 9,000 years.** (**a**) 9,000 to 4,200 years ago; (**b**) 4,200 to 1,300 years ago; (**c**) 1,300 years ago to the present day; (**d**) Legend and relative sea level (RSL) scenarios shown in (**a**)–(**c**) (see Fig. 5a). After deglaciation, during the Early Holocene Warm Optimum (EHO) (11.5–9.5 ka)[56], the land area available on Ardley Island was c. 0.6 km², c. 30–35% less than the present day. The eastern half of the island, where the current penguin colony is located, was bordered by steep cliffs forcing early to mid Holocene colonies to nest in the centre of Ardley Island. During the mid to late Holocene, relative sea level (RSL) fell, increasing the land area available. The amount of guano deposited in Ardley Lake declined after c. 1,300 cal a BP as some colonies relocated to the Lake Y2 and Lake G catchments[43]. The eastern side of the island became more easily accessible when RSL fell below 5 m above present sea level (m a.p.s.l.) after c. 1,300 years ago. Future colony population increases could be accommodated in the central area of Ardley Island. See Supplementary Fig. 1 and Supplementary Data 6 for modern-day penguin population data; see Supplementary Fig. 22 for an extended version of this figure. This figure includes material copyright of DigitalGlobe, Inc., All Rights Reserved, used with permission under a NERC-BAS educational license and not included in the Creative Commons license for the article.

BP. This 'neoglacial' is apparent in Peninsula-wide lake[58], marine[57,59] and ice core[29] records (Fig. 4f), and marked by the episodic re-advance of local glaciers[52,62] that slowed the rate of RSL decline on the SSI[63,64]. Although reconstructed temperatures in the YAN-GDGT record remained above average until c. 1.5 cal ka BP, sea-ice conditions in Maxwell Bay and the Bransfield Strait[52,57], and at some localities further south[65], became progressively less favourable between c. 2.3 and 1.8 cal ka BP (ref. 10), likely constraining the growth of the Ardley Lake and Lake Y2 colonies. The relatively minor (VEI < 3) T2 eruption event had a measurable impact on the Ardley Lake colony in the middle of guano-phase GP-5, c. 2.1 cal ka BP (Fig. 4b), but further deterioration in north-western AP climate into the 'neoglacial' minimum probably drove more progressive reductions in the Ardley Lake GP-5 and Lake Y2 colonies from c. 1.5 to 1.3 cal ka BP (ref. 43).

No significant changes in the Ardley Lake colony have occurred in the last c. 1,000 years (Fig. 4), although we detected a minor increase in guano input into Ardley Lake from c. 500 years onwards (Fig. 5), which coincides with the onset of warming detected in north-eastern AP lake[58] and ice core records[29] from c. 500 years, after the northern AP 'neoglacial' minimum. Elsewhere, increases in guano input into Lake Y2 on Ardley Island (Fig. 6) and in Hope Bay records from the north-eastern AP[10,16] have been linked to 'warming' associated with a 'Medieval Climate Anomaly'[43], inferred in some AP records between c. 1,200 and 600 years ago[48,56,66]. The increase in sub-fossil evidence of nesting sites across the AP in the last 1,000 years for all *Pygoscelis spp.*, regardless of whether they are 'ice-dependent' or 'ice-avoiding' species, could be a response to the more favourable 'Medieval Climate Anomaly' and/or post-'neoglacial' conditions, or reflect migratory shifts to newly emerged, low-lying coastal areas created by declining Holocene RSLs across the Peninsula[63,64,67–69] (for example, Fig. 5a, c). The absence of significant colonies at both the Ardley Lake and Lake

Y2 sites during the last c. 1,000 years mirrors the increased rate of RSL fall from c. 10 to 0 m a.p.s.l. (refs 45,64,70) (Fig. 6). This resulted in an increase in the available area for nesting sites near the coast and may have driven the relocation of penguins from the Ardley Lake catchment to the western side of Ardley Island, where much of the present day colony is located (Fig. 6c).

Although our new lake records suggest that warmer local atmospheric temperatures (Figs 4a,5) and the regional expression of the MHH led to increased penguin populations during guano phase GP-3 c. 4.5–4.3 cal ka BP and GP-4, c. 4.0–3.0 cal ka BP, our main finding is that large mid-late Holocene eruptions of the Deception Island volcano had a sporadic, but devastating, impact on the Ardley Lake penguin colony. All five phases of colony expansion occurred in the absence of large volcanic eruptions and three of the phases (GP-2, 3, 4) were abruptly ended by eruptions from the Deception Island volcano at c. 5.5–5.4 cal ka BP (T5a), 4.5–4.2 cal ka BP (T4) and 3.2–3.0 cal ka BP (T3b). Only guano phase GP-5 declined gradually, which we attribute to adverse 'neoglacial' conditions and migration away from nesting sites in the centre of the island towards newly emergent coastal areas. This guano phase appears to have been negatively impacted by the relatively minor T2 eruption event at c. 2.1 cal ka BP, but, in general, from c. 2,500 years onwards the Ardley Lake colony was too small to determine whether volcanic activity had a significant impact.

The remoteness of active volcanism in Antarctica means that modern-day analogues for volcanic impacts on penguin colonies are inevitably sparse and hard to quantify. Although the relative size of Deception Island eruptions is thought to have diminished in the late Holocene, comparatively small recent eruptions were locally hazardous, causing catastrophic flooding and lahars[20]. Larger mid-late Holocene eruptions are thought to have generated pyroclastic density flows and surges[20,21], which could have been particularly destructive at ground level. Circumstantial evidence from the volcanically active South Sandwich Islands

shows that, close to an erupting volcano, penguins are largely absent from areas affected by recent ash fall (Supplementary Fig. 23). Proximal fallout of coarse-grained tephra could lead to high localized mortality rates and significantly disrupt nesting activities, or drive migration to areas less affected by ash. The northern AP is defined as having a 'moderate to extremely high likelihood of supraglacial airfall by active volcanoes', where a 'significant risk to life' exists[31], mainly because the deposition of far-travelled fine black basaltic ash can lead to rapid snow-melt and (supraglacial) lahar formation[71]. Other indirect processes, such as burial by tephra, the generation of tsunamis, landscape destabilization, and the deposition of fine ash layers and aerosols could pose a substantial risk to distal penguin colonies[30], such as those on Ardley Island (Supplementary Fig. 12, Supplementary Discussion).

Physical disablement and poisoning are the most significant potential hazards to penguins, even though they are well-adapted to elevated levels of F, Cu, Cd, and Hg through high dietary intake[16]. Prolonged exposure to the fine abrasive glass particles in tephra while rearing chicks on land, or during feeding at sea, could adversely affect one (or all) of the major physiological processes of penguins and their prey (for example, respiration, digestion, immune function, vision; Supplementary Discussion). Colonies are more likely to be decimated if significant amounts of ash fell during the breeding season onto unhatched eggs or recently hatched chicks, or if one parent failed to return from foraging. Mature individuals on long forages, outside the areas directly affected by ashfall, would have the best chance of survival. Our recovery interval calculations suggest that these individuals may have been displaced to alternative nesting sites such as those that were occupied in the mid-southern AP at c. 5.4 and 4.2 cal ka BP (Fig. 5c). The physical landscape disturbance and presence of tephra over Ardley Island prevented sustainable post-eruption reoccupation of the site for, on average, 570 ± 110 years (weighted mean ± s.e.), but as much as c. 800–1,100 years following the largest mid-late Holocene (T5a) eruption c. 5,500 years ago (Table 2).

In conclusion, while we cannot assess some factors that determine breeding success such as pressure from predators[1], the ability to change diet[72], relative changes in species composition, or recent anthropogenic impacts[73], our detailed case study, using innovative methodologies, has revealed the key environmental factors influencing the Ardley Lake penguin colony through the Holocene. Once deglaciated, c. 8,500 years ago, climatic conditions on Ardley Island were sufficiently amenable throughout the rest of the Holocene for sustained penguin habitation. The Ardley Lake colony population maximum in first half of the late Holocene, c. 4,000 to 3,000 years ago, occurred during a period of particularly 'favourable' regional climate (sustainably warmer than the 6–0 ka average, with fewer storms) and local-regional sea-ice conditions (near-average 6–0 ka sea-ice cover/seasonal extent), which enhanced nesting and foraging success. Conversely, 'neoglacial' conditions and falling sea levels during the second half of the late Holocene (c. 2,000 years ago to present) contributed to colony decline and/or migration away from the centre of Ardley Island.

Overall, though, we found no consistent relationships between guano inputs into Ardley Lake and local-regional atmospheric and ocean temperatures or sea-ice conditions across the Holocene. Instead, the overriding driver of long-term penguin colony change on Ardley Island was volcanic activity from Deception Island. Three out of five mid-late Holocene guano-phases were interrupted by volcanic ash fallout from eruptions that were at least an order of magnitude greater than present-day eruptions. Mid-late Holocene volcanic eruptions deposited layer of volcanic ash across the landscape, which severely disrupted

nesting and foraging activities on Ardley Island. Moreover, the saw-toothed, asymmetric pattern of successive Ardley Lake penguin colony population maxima suggests that sustainable colony recovery following these mid-late Holocene volcanic eruptions was slow, taking, on average, between 400 and 800 years, but possibly as much as 1,100 years during the most disruptive phase of volcanism (T5) c. 5.5–4.6 cal ka BP.

## Methods

**Sample collection and laboratory analyses.** Sediment cores extracted from Ardley Lake (ARD) and Yanou Lake (YAN) were analysed for the concentration of elements associated with changing inputs of penguin guano and volcanic ash deposits using complementary multi-proxy biogeochemical techniques described in this section and in Supplementary Methods.

Chronologies for the lake sediment cores were established using 18 (ARD) and 15 (YAN) AMS radiocarbon ([14]C) ages from, in order of preference: (1) moss macrofossil layers (consisting of hand-picked fine strands of the aquatic moss *Drepanocladus longifolius* (Mitt.) Paris, but also occasional layers of *Campylium polygamum* (Schimp.) Lange & C.E.O. Jensen, and some unidentifiable/mixed species moss fragments—considered more likely to have been reworked; (2) terrestrial and/or lacustrine algae; (3) other intact macrofossils and sub-fossils, including bones (bone-collagen, where extractable); (4) other (macro)fossil fragments; (5) organic-rich bulk sediments and, near the base of each core and as a last resort, (6) bulk glaciolacustrine or glaciomarine sediments (Supplementary Table 2). Bulk sediments were only dated where macrofossils were absent, while paired macrofossil or bone-collagen and bulk sediment samples were measured in the surface sediment and wherever present to check for any systematic offsets between ages obtained from different carbon sources (see Supplementary Methods, Supplementary Notes 5 and 6 for more details).

Measured radiocarbon ages from samples shown in Supplementary Table 2 were calibrated using SH13 and MARINE13 calibration curves and age-depth models were generated using Bayesian age-depth modelling techniques (Fig. 2, Supplementary Figs 3,9, Supplementary Methods and Supplementary Notes 5–7). All the 'as measured' (uncalibrated) radiocarbon age data shown in Supplementary Table 2 were used as input data for final age-depth model runs (ARD-M5, YAN-M4, ANVERS-M1, where model run number is indicated by the -M suffix). The weighted mean basal age of the ARD core was 8,750 cal a BP [8,410–9,230 min –max. 95% confidence age range]. Error analysis shows that whole record mean 95% confidence age-depth model uncertainties, rounded to the nearest 10 years with 95% confidence minimum to maximum age ranges (in years at depths in cm) shown in square brackets are: ARD-M5: 410 years [0.6 years at 0 cm–1,430 years at 335 cm], YAN-M4: 600 years [160 years at 7.7 cm–1,990 years at 340 cm], and ANVERS-M1: 1,420 years [590 years at 0 cm–1,810 years at 375 cm]. Equivalent values for the late Holocene in each record are: ARD-M5: 640 years [0.6 years at 0 cm–750 years at 20 cm], YAN-M4: 680 years [160 years at 7.7 cm–760 years at 2.7 cm] and ANVERS-M1: 1,370 years [590 years at 0 cm–1,600 years at 155 cm] (Supplementary Note 7).

Sub-samples for carbon, nitrogen, XRF and ICP-MS (quantitative, dry mass) bio-element analyses were taken at 1 cm intervals from ARD and YAN lake sediment cores, lyophilized and ground with an agate ball mill and manually with an agate pestle and mortar. ARD samples were analysed for total carbon (TC), total sulphur (TS) and total nitrogen (TN) using a CNS analyser (vario EL Cube, Elementar, Germany) equipped with a solid-state infrared and a heat conductivity detector. TC and TS measurements were conducted on YAN sediments by means of an ELTRA CS analyser. For data replication and comparison purposes, the total inorganic carbon (TIC) of 69 (out of 385) samples from both cores was determined coulometrically by a CM 5012 $CO_2$ coulometer coupled to a CM 5130 acidification module (UIC, USA) while total organic carbon was then calculated as the difference between TC and TIC (TOC% = TC% − TIC%). Owing to the high correlation between TC and TOC ($R^2 > 0.9997$) and the negligible TIC concentrations (av. 0.10 and < 0.01 mass%) compared to TC values (av. 6.2 and 2.5 mass%) in both cores, TC is considered to reflect the amount of organic carbon in both lake sediments. Quantitative XRF analysis for major and trace elements ($SiO_2$, $Al_2O_3$, CaO, $K_2O$, $Na_2O$, $P_2O_5$, As, Ba, Cu, Co, Ni, Sr, Y, Zn, Zr) was carried out with a conventional wavelength dispersive X-ray fluorescence (WD-XRF) spectrometer (Philips PW 2400). For WD-XRF, glass beads were prepared following standard procedures[74] and measurements undertaken in random order to avoid artificial trends. Trace element analysis (Cd, REE) of selected samples was performed by Inductively Coupled Plasma Mass Spectrometry (ICP-MS, Element 2 mass spectrometer, Thermo Scientific, Germany) at 2,500-fold dilution. For additional details concerning ICP-MS measuring conditions, see Schnetger[75]. Selenium was determined on acid digestions by graphite atomic absorption spectrometry (G-AAS) using a Unicam 939 QZ AA spectrometer and a Zeeman-effect background correction. A Milestone DMA-80 Direct Mercury Analyser was used for the measurement of mercury via cold vapour atomic absorption spectroscopy (CV-AAS). ICP-MS and G-AAS standard acid digestion procedures are described in Supplementary Methods.

Following established procedures, we used visual descriptions, smear slides and contiguous micro-XRF (μ-XRF) scanning, at 200 μm and 2 mm intervals, to

determine the precise position of tephra deposits (see Supplementary Methods, Supplementary Note 9, Supplementary Figs 3,7,10,12 for details). Electron Probe Micro-Analysis (EPMA) of 165 glass-shards from the most prominent T4–T7 tephra layers in the YAN, ARD and Beak 1 Lake records (Beak Island; Layers $T_{a–e}$ (ref. 63)) was used to determine the eruption characteristics and source of mid-late Holocene eruptions that likely had the biggest environmental impacts. Results were compared with glass-shard analyses from age-equivalent tephra layers in marine cores PC460/461 from the Scotia Sea (new data; this study) and similarly analysed data in Moreton and Smellie[23] from the Scotia Sea, northern Weddell Sea and Boyd Strait, and in Fretzdorff and Smellie[22] from the Bransfield Basin (see Supplementary Methods, Figs 20,21, Supplementary Table 10).

For the YAN-GDGT reconstructed temperature record, temperature-sensitive glycerol dialkyl glycerol tetraethers (GDGTs) biomarkers were extracted from 41 samples using a microwave-assisted solvent extraction[76,77]. Contiguous 1 cm subsamples were taken in the top 20 cm of the YAN record, and at 2 cm intervals between 20 and 38 cm and 188 and 210 cm core depth (dated to c. 6.1 cal ka BP). Freeze dried and homogenized sediment samples weighing 0.2–4.3 g were microwave-extracted in DCM:Methanol (3:1, v/v). The total extracts were saponified and GDGTs isolated following ref. 76. Prior to analysis the GDGT extracts were filtered through a 0.2 μm Whatman PTFE filter. GDGT analysis was undertaken using an Acquity Xevo TQ-S (triple quadruple with step wave; Waters Ltd.) LC-MS set up with an atmospheric pressure chemical ionization source (Ion saber II) operated in positive ion mode. Analytical separation was achieved using a Grace Prevail Cyano HPLC column (3 μm, 150 × 2.1 mm i.d.) fitted with an in-line filter (Waters Acquity UPLC in-line filter, 0.2 μm) at 40 °C using a binary solvent gradient where eluent A was hexane and eluent B was propanol. The flow rate of the mobile phase was 0.2 ml per minute with a gradient profile of 99% A 1% B (0–50 min); 98.2% A 1.8% B (50–55 min); 90% A 10% B (55–65 min) and finally 99% A 1% B (66–80 min). The LC-MS settings were: source offset 50 V, capillary 1.5 kV, desolvation temperature 200 °C, cone voltage 30 V, desolvation gas ($N_2$). Detection was achieved using selected ion monitoring of targeted $[M + H]^+$ ions (dwell time 50 ms). The target ions were $m/z$ 1,302, 1,300, 1,298, 1,296 and 1,292 for the isoprenoid GDGTs (isoGDGTs) compounds and 1,050, 1,048, 1,046, 1,036, 1,034, 1,032, 1,022, 1,020 and 1,018 for the branched GDGT (brGDGTs) compounds. GDGTs were identified and integrated using MassLynx software (v.4.1) and GDGT-derived temperatures were calculated using the Antarctic and sub-Antarctic GDGT-temperature calibration[77] (see Data Analysis section). The Antarctic and sub-Antarctic GDGT-MSAT surface calibration data set (comprising 32 sites in total) includes surface sediments from Yanou Lake, Ardley Lake and three other lakes from Fildes Peninsula, along with two further lakes from Potter Peninsula and four lakes from the Trinity Peninsula, north-eastern AP (including Beak Island).

Diatom analysis of the ARD and YAN records is as described in ref. 45 and summarized in Supplementary Fig. 14, while Anvers Shelf marine core (ANVERS) diatom analysis and chronology are described in Supplementary Methods and summarized in Supplementary Fig. 15.

**Data analysis.** Bio-element assemblages are immobile in (Antarctic) lake sediments[43] and guano input to sediments leads to the formation and preservation of stable phosphates, such as struvite ($Mg(NH_4)PO_4$ x 6 $H_2O$), leukoposphite and, in particular, hydroxylapatite ($Ca_5(PO_4)_3(OH)$), which is one of the dominant compounds found in ornithogenic soils/sediments on King George Island. During the precipitation of apatites, an exchange between $Ca^{2+}$, $PO_4^{3-}$, $F^-$ and $OH^-$ and elements such as Ag, Br, Ba, Cd, Cu, Cr, I, Na, Mg, Mo, Pb, S, Se, Sr, U, V, Y and Zn is possible, coupled to microbial-mediated degradation of solid phases[78] (see Supplementary Note 1 for more details). These trace elements, which have naturally higher concentrations in penguin guano, are enriched in these phosphate phases and then immobilized in soils and sediment profiles during this substitution process. While the high correlation between Mg and P in ornithogenic soils from Vestfold Hills has been used to indicate the presence of struvite[79], their lack of correlation in the Ardley Lake sediments means that Mg is likely derived from the bedrock lithology (Supplementary Fig. 3, Supplementary Table 4), and hydroxylapatite ($Ca_5(PO_4)_3(OH)$) formation reflects guano input (Supplementary Fig. 6).

The average relative proportion of guano or ornithogenically derived sediment in the Ardley Lake sediment matrix ($F_{o.sed.}$) was estimated by using the mixing equation (equation (1)), modified after Shultz and Calder[80]:

$$F_{o.sed.}(rel.\%) = \frac{\sum_{El=Cu,Sr,Zn,P}\left(\frac{(El/Al)_{smp} - (El/Al)_{bgd}}{(El/Al)_{o.sed.} - (El/Al)_{bgd}}\right)}{4} \times 100 \quad (1)$$

$(El/Al)_{smp}$ is the element/aluminium ratio of selected bio-elements (Cu, P, Sr, Zn) in Ardley Lake sediments and $(El/Al)_{bgd}$ (mean Ardley Island bedrock[51]) and $(El/Al)_{o.sed.}$ (guano-bearing ornithogenic sediment and soils[49]) represent the respective ratios in both end members. In order to minimize misinterpretation of this proxy, we calculated the average fraction of ornithogenic sediment and soils in the lake sediments using a combination of four chemically unrelated El/Al ratios. This provided a buffering effect on possible variations in ornithogenic soil and bedrock composition over time. In the resultant $F_{o.sed.}$ percentage plotted in Fig. 4b, for example, $F_{o.sed.} = 20\%$, means that 20% of the sediment sample is eroded ornithogenic sediment-soil and 80% eroded bedrock. Theoretically, all $F_{o.sed.}$ values

> 0% provide evidence for guano input above the local bedrock-terrigenous background level, but we used a weighted mean $F_{o.sed.}$ value of >10% at the cut-off for CONISS-defined geochemical zone to indicate the sustained presence of penguins around Ardley Lake (Table 1, Supplementary Table 7). We calculated post-eruption colony recovery intervals as the time taken until the percentage $F_{o.sed.}$ value first returned to >10% within the subsequent guano phase to represent the return of a sustained penguin presence around Ardley Lake. Measurement errors of <1% mean our definitions provide very conservative buffers of penguin presence during guano phases 1–5 (GP-1 to GP-5) (Table 2).

Penguin population modelling results shown in Fig. 5 utilized dry mass $F_{o.sed.}$·accumulation rates ($F_{o.sed.}$ DMAR in Fig. 4b), which were calculated following standard procedures[81] (that is, multiplying $F_{o.sed.}$ percentages by the dry mass accumulation rate, calculated from sedimentation rate and dry mass density data). We also used published penguin population parameters[40,41] and the 'most-realistic' sediment focussed ellipsoid basin accumulation models[81], assuming a <30 ° slope-angle accumulation area within Ardley Lake of 5,682 $m^2$ (out of a maximum 7,274 $m^2$ lake basin area) (see Supplementary Fig. 2). Further accumulation area scenarios, equations and references can be found in Supplementary Note 8. We considered the shape of Ardley Lake basin to approximate between a steep-sided hyperboloid basin and an ellipsoid basin (as below).

To summarize, the basin-wide guano influx rate, $S_G$, at time $t$, and total Ardley basin maximum core depths of $z$ (water depth + sediment depth, in cm) at time $t$, was calculated as follows:

For an ellipsoid-basin:

$$S_G(t)_{ellip} = -\frac{2}{3}\left[1 - \frac{z_a^2}{(2z_a + z)^2}\right]\left(\frac{dz_G}{dt_G}\right) \quad Units: g\,cm^{-2}a^{-1} \quad (2)$$

and

$$z_a = z_m[2 - 3(\bar{z}:z_m)_0]/3[2(\bar{z}:z_m)_0] - 1 \quad Units: cm \quad (3)$$

For a hyperboloid basin shape:

$$S_G(t)_{hyper} = -\frac{2}{3}\left[1 - \frac{z_m^2}{(2z_m - z)^2}\right]\left(\frac{dz_G}{dt_G}\right) \quad Units: g\,cm^{-2}a^{-1} \quad (4)$$

where $z$ = core depth at time $t$; $z_m$ = maximum depth of the original basin (that is, current lake depth + sediment record depth); $(\bar{z}:z_m)_0$ = ratio of the mean depth $(\bar{z})$ to the maximum depth of the original basin $(z_m)$; $(dz_G/dt_G)$ = guano dry mass accumulation rate = $F_{o.sed}$ DMAR in $g\,cm^{-2}a^{-1}$.

Assuming Ardley Lake best approximates to an ellipsoid basin shape (that is, the 'more likely' basin shape scenario), the total basin-wide guano accumulated ($I_G$) over the 'more likely' <30 degree slope accumulation area ($A_{<30}$) within Ardley Lake (Supplementary Fig. 2), corrected for the effects of sediment focussing, is then:

$$I_{G-A<30-ellip}(t) = S_G(t)_{ellip}.[A_{<30}] \quad Units: g\,cm^{-2}a^{-1}cm^2 = g\,a^{-1} \quad (5)$$

where $A_{<30}$ = 5,682 ± 568 $m^2$ = 5.68×$10^7$$cm^2$; $S_G(t)_{ellip}$ = the value for each 100-year interval obtained from equation (2).

Using guano production rates of 84.5 ± 21.1 g (25% error estimate applied) per penguin per day on Ardley Island, and a mean density of 0.31 ± 0.19 gentoo penguins per $m^2$ (for Signy Island)[40,41], we estimated the total amount of guano delivered by erosion from the catchment into Ardley Lake as the guano yield ($G_y$) and the area required for the reconstructed population. As the main breeding season and ice/snow-free periods, when active erosion from catchments to lakes occurs on Ardley Island, last for approximately 3–4 months each year (91.3 ± 22.8 days; 25% error estimate; equation (6) and Supplementary Table 9), we calculated the maximum amount of guano deposited per penguin ($p^{-1}$), per year ($a^{-1}$) inside the catchment and delivered to the lake as the maximum possible guano yield ($G_{y-100\%}$) as follows:

$$G_{y-100\%} = \bar{D} \cdot P_g = 7,716 ± 2,728\,g\,a^{-1}p^{-1}(c.35\% \text{ combined error}) \quad (6)$$

where $\bar{D}$ = mean occupation time per year ± estimated error = 365.25/4 or 91.3 ± 22.8 days (25% error applied); $P_g$ = amount, in grams (g) of guano deposited per year ($a^{-1}$) per penguin ($p^{-1}$).

Several studies have shown that catchment erosion and deposition in small closed-basins with a single catchment and non-complex inflow characteristics (for example, Ardley Lake) can be approximated by a linear relationship (see Supplementary Note 8 for references); thus, a reasonable approximation of the total amount of sediment or, in this case, guano delivered ($G_D$), by erosion ($G_e$), into the Ardley Lake basin, can be obtained, simply from ratio of lake-area to catchment-area, expressed as an estimated percentage:

$$G_D = \frac{G_{y-100\%}}{G_e} = \frac{A_{lake}}{A_{catch}} = \frac{7,274}{66,249}\frac{m^2}{m^2} = 1 : 9 = 11\% \quad (7)$$

where $G_D$ is the ratio, expressed as a percentage, of the total guano produced in the catchment ($G_{y-100\%}$) and the total guano-eroded from the catchment ($G_e$) into the lake, which, here we approximate to the lake area: lake catchment ratio, $A_{lake}/A_{catch}$.

For Ardley Lake, the amount of guano delivered in grams (g) per year ($a^{-1}$) per penguin ($p^{-1}$) in the 11% guano ($G_{y-11\%}$) yield scenario is given by:

$$G_{y-11\%} = 0.11(\bar{D}P_g) = 847 \pm 300 \, \text{g} \, a^{-1}p^{-1} \quad (8)$$

However, small lakes with restricted inflow and/or low rates of catchment erosion or slow sedimentation rates (that is, most typically high latitude, frozen lakes, for example, Yanou Lake during the late Holocene) are better represented by an exponential sediment delivery model, $G_{D-ST}$ (ref. 81), which, for Ardley Lake, produces $G_{D-ST} = 0.96\%$ as follows:

$$G_{D-ST} = 0.36(A_{max}^{-0.2}) = 0.36(7,274^{-0.2}) = 0.96\%. \quad (9)$$

This allowed us to estimate the number of penguins ($P$) present in the catchment at time ($t$) for three guano-yield scenarios, $G_{y-100\%,-11\%,-0.96\%}$:

$$G_{y-100\%} = 7,716 \pm 2,728 \, \text{g} \, a^{-1}p^{-1}$$

$$G_{y-11\%} = 847 \pm 300 \, \text{g} \, a^{-1}p^{-1}$$

$$G_{y-0.96\%} = 74 \pm 26 \, \text{g} \, a^{-1}p^{-1}$$

as follows

$$P_{A<30-ellip}(t) = \frac{I_{G-A<30-ellip}(t)}{G_{y-100\%,-11\%,-0.96\%}} \quad \text{Units}: \frac{\text{g} \, a^{-1}}{\text{g} \, a^{-1}p^{-1}}$$
$$= P(\text{number of penguins}) \quad (10)$$

Since the catchment colony would have produced and/or deposited guano outside the catchment area (for example, on foraging trips), and not all guano deposited in catchment areas is currently eroded into lake due to various factors, for example, utilization by vegetation, we consider the $G_{y-100\%}$ to be very unlikely. Therefore, we consider the $G_{y-11\%}$ and $I_{G-A<30-ellip}(t)$ to best represent a possible 'most likely' minimum population scenario as follows:

$$MINP_{A<30-ellip}(t) = \frac{I_{G-A<30-ellip}(t)}{G_{y-11\%}} \quad \text{Units}: \frac{\text{g} \, a^{-1}}{\text{g} \, a^{-1}p^{-1}}$$
$$= P(\text{number of penguins}) \quad (11)$$

As a reality check, the $G_{y-11\%}$ and $I_{G-A<30-ellip}(t)$ scenario recreated the near-absence of a penguin colony ($<10$ penguins) in the modern day catchment. Conversely, we consider that the $G_{y-0.96\%}$ and $I_{G-A<30-ellip}(t)$ scenario represents the 'most likely' maximum population scenario:

$$MAXP_{A<30-ellip}(t) = \frac{I_{G-A<30-ellip}(t)}{G_{y-0.96\%}} \quad \text{Units}: \frac{\text{g} \, a^{-1}}{\text{g} \, a^{-1}p^{-1}}$$
$$= P(\text{number of penguins}) \quad (12)$$

These two end-member scenarios are shown in Fig. 5, but since sedimentation rates in Ardley Lake have varied significantly through time, it is possible that, when erosion rates are high, for example, between c. 5.5 and 4.5 cal ka BP, the Ardley Island colony population would be better estimated by a guano-yield scenario somewhere between 11 and 100%. To further validate our findings, we also calculated the catchment area required by the reconstructed colony population using published gentoo penguin density values[40] and calculated mixed-species density values, partitioned according to published modern-day Ardley Island gentoo-Adélie-Chinstrap species compositions[41] and the corresponding species density values[40] (Supplementary Fig. 19b, Supplementary Table 9).

The 'most-likely' minimum guano-delivery (yield) scenario assumes that 11% of the guano produced by penguins in the Ardley Lake catchment was eroded into Ardley Lake. This scenario produced a peak guano phase GP-4 colony population of $1,024 \pm 574$ penguins, which occupied an area of $1,679–3,303 \, \text{m}^2$ out of the $66,249 \, \text{m}^2$ Ardley Lake catchment area (based on partitioned mixed species minimum density–gentoo penguin maximum density values[40]). Meanwhile, the 0.96% guano-yield scenario produced a maximum peak GP-4 colony population of $11,723 \pm 5,104$, which corresponds to an occupied area of $20,686–37,815 \, \text{m}^2$ (Supplementary Fig. 19). Therefore, the c. $66,250 \, \text{m}^2$ Ardley Lake catchment area is easily large enough to accommodate all modelled population scenarios shown in Fig. 5. Assuming a hyperboloid basin shape for Ardley Lake would result in an average $33 \pm 25\%$ reduction in the reconstructed population across all guano delivery scenarios.

It is important to note that the apparent increase in the amount of guano deposited into Ardley Lake over the last 500 years is less than the significant 10% $F_{o.sed.}$ level we used to determine colony presence elsewhere in its Holocene record, and only equates to a total of $65 \pm 48$ penguins (using the 11% guano-delivery scenario). This also illustrates why the Ardley Lake guano phase colony was too small after c. 2,500 years for us to determine whether volcanic activity had a significant impact.

Correlation analysis of geochemical data was undertaken using R 2.15.2 (R Foundation for Statistical Computing). Hierarchical R-mode cluster analyses were conducted with the R package 'Pvclust' (version 1.2-2) using average linkage and a correlation-based dissimilarity matrix (see Supplementary Methods for software references). Based on multi-scale bootstrap resampling (number of bootstraps: 10,000), approximately unbiased $P$ values were further calculated to assess the uncertainties in cluster analysis. We used constrained incremental sum of squares (CONISS) stratigraphically constrained cluster analysis with broken stick analysis to define Diatom Zones (D) in the ARD, YAN and ANVERS records, Geochemical Zones (GZ) in the ARD and YAN records, and Guano phase GP-1 to GP-5 boundaries in the ARD record. Downcore cluster-zonation was undertaken on ecologically grouped diatom percentage data and square-root transformed geochemical data using the R packages vegan and rioja (see Supplementary Methods for software references). Downcore trends in diatom and geochemical data sets using Principal Components Analysis (PCA) were undertaken using square-root transformed percentage XRF data and µ-XRF (Total Scatter Normalised (TSN)) percentage geochemical data and percentage diatom count data.

For climate and sea-ice comparisons, the Yanou Lake reconstructed GDGT temperature (YAN-GDGT) data (grey plot, with RMSE errors shaded grey in Supplementary Fig. 16a) and the Maxwell Bay measured TOC data (sea-ice proxy record)[52] (grey plot in Fig. 4c) were smoothed using polynomial negative exponential regression analysis. Results were similar to smoothing by LOESS first-order polynomial regression with tri-cube weighting (not shown). Reconstructed GDGT temperatures reflect MSAT (December, January, February or DJF) in the shallow and well-mixed open lake water environment of Yanou Lake[76,77]. For the YAN record, whole YAN-GDGT data set 5th and 95th percentile outliers ($<-0.33\,°C$ and $>8.61\,°C$) $11.34\,°C$ and $13.84\,°C$ at $1,164$ cal a BP and $3,135$ cal a BP, respectively, were excluded by smoothing analysis (Fig. 4a,c, Supplementary Fig. 17a). These values were also greater than the $10.3\,°C$ maximum limit of the surface sediment-MSAT Antarctic and sub-Antarctic GDGT-MSAT calibration (ANT-GDGT) data set[77]. All reconstructed GDGT temperatures were greater than the $-2.2\,°C$ minimum limit of the ANT-GDGT calibration data set. The YAN-GDGT $<10\,°C$ data set was used in 6–0 ka weighted mean temperature anomaly calculations, statistical analyses and hypothesis testing. The mean $\pm 1\sigma$ 6–0 ka reconstructed YAN-GDGT $<10\,°C$ data set weighted mean MSAT value of $+2.39 \pm 2.67\,°C$ [95% confidence interval: 1.52–3.25 °C] (Bootstrapping, $n = 10,000$, $+2.38 \pm 2.61\,°C$ [1.52–3.25 °C]; 5th/95th percentile $= -0.51/ +7.96\,°C$; weighted by Antarctic GDGT calibration data set RMSE value of $1.45\,°C$; Supplementary Table 7) encompasses the modern-day (1968–2015 CE) observational mean summer (DJF) temperature from Bellingshausen Research Station on Fildes Peninsula of $+1.09 \pm 0.56\,°C$ ($2\sigma$ 95% measured range $= -0.03$ to 2.21 °C; $n = 48$, 1968–2015 CE, where year refers to December) (Marshall, pers. comm.; SCAR-READER database: http://www.antarctica.ac.uk/met/READER/). On a regional to global-scale modern-day reconstructed lake sediment GDGT-temperatures broadly reflect changes in air temperature[77]. At the local, small lake scale, water temperature might decouple from observed MSAT due to factors such as the thickness and length of seasonal lake ice-cover and the volume of meltwater input, both of which suppress lake water temperatures. Conversely, since water bodies store excess heat generated during extended ice-free seasons, small and/or shallow lakes ($<10$ m deep) reach temperatures of 6 °C or more (for example, Sombre Lake, Signy Island)[82]. As the focus of this paper is penguin colony reconstruction, we investigate these aspects further in a forthcoming paper.

After smoothing, all climate (YAN-GDGT, JRI[29] and PD[48]); and sea-ice (Anvers Shelf and Maxwell Bay[52]) data sets were resampled at 100-year intervals (see Supplementary Methods). We then used $>6–0$ ka upper bound on the weighted mean (95% confidence level) temperature, temperature anomaly, diatom open water ratio and smoothed TOC- and diatom-based sea-ice proxy data (Fig. 4) to define 'warm' and 'sea-ice free' (open-water) periods. We chose the 6–0 ka period and values greater than the upper bound on the weighted mean at 95% confidence level of the weighted mean as this provides the best longer-term assessment of the mean temperature and variability that penguin colonies on Ardley Island would have encountered. It also corresponds to the time period covered by the terrestrial Yanou Lake GDGT-MSAT record. Using the Pre-industrial era (1,000–250 cal a BP) and its upper bound on the weighted mean value (95% confidence) produced similar timings for 'warm' and 'open-water' intervals.

To compare variations in cross-Peninsula climate during guano and non-guano phases, we undertook normality tests and produced a set of descriptive statistics for the whole Ardley $F_{o.sed.}$ data set (weighted by $1\sigma$ error), the YAN-GDGT $<10\,°C$ temperature data set, the JRI temperature anomaly data set[29], and the PD 0–200 m sea-surface temperature (PD-SST) record[48] (weighted by published errors, where available). Using $\alpha = 5\%$ and 10% significance levels for three hypothesis test scenarios A–C shown in Supplementary Table 8, we first performed Fisher's F-test Variance Ratio analysis. Since some data sets were not normally distributed, we then used the Mann–Whitney $U$-statistic to test the hypothesis that temperature and temperature anomaly distributions in the YAN-GDGT $<10\,°C$, JRI, PD-SST records were statistically different during the five guano-influenced phases (Scenario 1A in Supplementary Table 8) compared to the eight non-guano phases (Scenario 2A). We then repeated this process comparing guano phases of the mid-Holocene (8.2–4.2 ka; 1B) to those of the late Holocene (4.2–0 ka; 2B), and for late Holocene guano phases (1C) versus non-guano phases (2C). We also undertook similar hypothesis testing using the Maxwell Bay sea-ice and Anvers Shelf open-water data sets to determine whether sea-ice conditions within guano and non-guano were statistically different in the three scenarios (A–C) described above. Results are summarized in Supplementary Table 8.

Apart from one recent minor eruption, all visible (that is, >2 mm thick) Holocene-age tephra deposits in lake, marine and ice core records of the northern Antarctic Peninsula (N-AP), SSI and Scotia Sea have been linked to VEI = 3 or VEI > 3 eruptions from Deception Island[20–23,27–29,58,63,83,84] (Supplementary Note 9, Supplementary Figs 20, 21, Supplementary Tables 3, 10). Tephra from even relatively small 1967 and 1970 CE Deception Island eruptions (VEI = 3) reached more than 150 km (ref. 22) and is present in the JRI ice core[29]. Neither the 1976 or 1970 eruptions formed visible ash layers near the top of the ARD core, probably because only up to 1 mm of ash was deposited on Fildes Peninsula[21]. Therefore, we broadly equated major VEI = 3 eruptions from Deception to visible tephra deposits <1 cm thick in the YAN and ARD records, and airfall ash deposits >1 cm thick to VEI > 3 eruptions. Since the post-deglaciation (after c. 8–7.5 cal ka BP) lithogenic deposition style in YAN and ARD is predominantly fine-grained silt-clay, the sand dry mass accumulation rate (DMAR) provided a useful proxy for eruption size and the volume of tephra deposited onto Ardley Island and King George Island during VEI > 3 eruptions (Fig. 4a,b, Supplementary Figs 12,20,21), while elevated Ca/Ti ratios and PCA2 values from combined 2 mm and 200 µm µ-XRF scanning data and the reliable single-species aquatic moss age-depth model provided precise stratigraphic constraints on the positions (and ages) of tephra deposits in the YAN record (Supplementary Note 9, Supplementary Figs 9–11,16, Supplementary Table 3).

**Data availability.** Key data sets for this study are in Supplementary Data files 1–7. All data produced by this study and associated IMCOAST/IMCONET projects are available from the corresponding author (sjro@bas.ac.uk) and the NERC Polar Data Centre (BAS) (https://www.bas.ac.uk/data/uk-pdc/) and Pangaea Data Publisher website (AWI) (https://www.pangaea.de/). Antarctic location maps and spatial data in Fig. 1 are the LIMA (Landsat Image Mosaic of Antarctica) downloaded from https://lima.usgs.gov/index.php and the SCAR Antarctic Digital Database http://www.add.scar.org/, used under a Creative Commons license. The basemap in Fig. 1a and Supplementary Figs 20,23 is the ESRI Ocean Basemap (http://goto.arcgisonline.com/maps/Ocean_Basemap) used under a BAS-NERC software license. Satellite images and image derivatives in Figs 1c,d and 6 and Supplementary Figs 2,22,23 are copyright of DigitalGlobe, Inc. All Rights Reserved (Catalogue numbers: 1030010020C0C900 and 11MAR04115702), reproduced at low resolution in a static format under a BAS-NERC educational license, and are not included in the Creative Commons license for the article.

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

## Acknowledgements

This study forms part of the ESF-funded IMCOAST project, AP-6 led by S.J.R., and the EU project IMCONet (FP7404 IRSES, action no. 319718; coordinated by the Alfred Wegener Institute, Helmholtz Centre for Polar and Marine Research, Germany) with additional funding from the Natural Environmental Research Council (NERC), and the German Research Foundation (DFG project no. BR 775/25-1) and logistic support from the NERC-British Antarctic Survey (BAS), HMS *Endurance* and 892 Naval Air Squadron, the Alfred Wegner Institute (AWI) and the Instituto Antártico Argentino (IAA). We thank NERC for funding studentship NE/J500173/1 (BAS and Newcastle University) to L.C.F. (supervisors S.J.R., E.J.P., D.A.H., S.J.). We thank colleagues at the Chinese Great Wall Station and Russian Bellingshausen Bases on Fildes Peninsula for their generous hospitality, H. Biester and T. Riedel from Braunschweig University for help with Hg analyses, technical assistants at the Institute for Chemistry and Biology of the Marine Environment (ICBM), Hilary Blagbrough and Paul Geissler at BAS, S. Xu at the SUERC AMS Facility, Chris Hayward (EPMA, University of Edinburgh), the Natural History Museum (Tring) for access to, and advice about, their avian fossil-bone collection, Helen Talbot and Frances Sidgwick for running samples on the LC-MS at Newcastle University, and Huw Griffiths (BAS) for assistance with ARC-GIS. We thank Phil Trathan, Claire Waluda and three reviewers for their constructive comments.

## Author contributions

S.J.R., P.M., L.C.F., D.A.H. and M.J.B wrote the paper. Fieldwork was carried out by S.J.R., D.A.H., E.P.H., M.J.B. and P.F.; P.M., S.J.R., L.C.F., E.J.P., S.J., J.L., B.S., H.-J.B. undertook analytical work and data analysis and contributed to the manuscript; S.J.R. and S.J.D. undertook and interpreted core scan data, A.R.H. and C.S.A. collected and conducted diatom analyses on marine cores from the Anvers Shelf; L.I. and P.F. performed satellite vegetation mapping analysis; R.O. identified mosses used for radiocarbon dating; S.G.M. undertook SUERC radiocarbon dating analyses and provided his previously analysed Deception Island EPMA data. All authors contributed to the interpretations and commented on the manuscript.

## Additional information

**Competing interests:** The authors declare no competing financial interests.

