## [Peer Review File · Nature Communications]

Reviewers' comments:

Reviewer #1 (Remarks to the Author):

This paper provides an extensive analysis of lake sediments from a core collected on Ardley Island, King George Island, Antarctic Peninsula. The authors identify sequences of bio-elements in the core that are characteristic of penguin guano to infer occupation of the island by breeding penguins during different time periods over the past 8000 yrs. They further posit that volcanic activities have curtailed or prevented occupation of this area during periods when no bio-elements are present in the sediments. Instead, a negative correlation between tephra deposits and the bio-elements suggests that volcanic eruptions have caused abandonment of the island over millennia. Their data also provide the oldest evidence for penguin occupation in the northern Antarctic Peninsula. The inclusion of a second core from a control lake (Yanou Lake), situated far from any penguin influence, is an excellent addition to the study and quite useful in verifying the bio-element input from guano at Ardley Island as well as the negative correlation with the tephra. The geochemical and statistical analyses are sound and appropriate and the resulting patterns they found in relation to volcanic events are compelling and novel. There are a few places where the paper could use more clarification and detail, and where the authors could address discrepancies in the data from their and other studies at Ardley Island and the northern Antarctic Peninsula.

There are some key citations that were missed and a few corrections that are needed. First, the authors refer to ancient penguin remains throughout the paper as 'archaeological' but this term refers strictly to human artifacts and remains. While archaeological methods have been used to excavate ancient penguin sites, the remains here should be referred to as 'sub-fossil' and not archaeological. On line 50, the scientific name for Adélie penguin does not have the accent mark above the "e". They should also avoid using the term 'rookery' (Line 60) as that is an old term for a breeding colony that originates from rook colonies in Europe. They should instead consistently refer to penguin breeding sites as colonies. In addition, the Chinese research on Ardley Island is well cited by the authors, but they have published at least two other papers (Sun et al. 2001, Chinese Journal of Polar Science and Wang et al. 2007, Polar Biology) on bio-elements from a sediment core from Ardley Island (which they refer to as sample Y2 from Antarctic West Lake, just west of Ardley Lake). The authors have already compared their data to other published data from this Y2 core (their citations 18, 20, and 41), but should also look at these papers, especially the more recent one, and the conclusions drawn by the authors on the Gentoo occupation. In addition, another study that reports mid-Holocene penguin remains from King George Island is by del Valle et al. (2002, Antarctic Science) and a more recent study on the age of the current Gentoo penguin colony on Ardley Island, which extends to 800-1000 BP, is in Emslie et al. (2013, Antarctic Science). Finally, the authors indicate that three juvenile penguin bones were radiocarbon dated from their lake core (supplementary Fig. 4), but these dates are not included in supplementary Table 2 or reported anywhere else in the manuscript that I can find. Since these are the only direct dates on penguin remains from their lake core, and thus the penguin occupation, it is frustrating to not know the dates obtained. These bone dates also could help verify the age of the sediment core based on the other dates provided in Table 2. Overall, a more comprehensive summary of the new data presented by the authors with these additional data will produce a well-rounded and excellent contribution to the literature on penguin occupations in the northern Antarctic Peninsula.

Another topic worthy of additional discussion is that, while Ardley Lake is not too far from an active Gentoo penguin colony on the east end of the island, the area around the lake itself has shallow rocky soil, shallow bedrock, and no evidence for abandoned penguin colonies anywhere near the lake. Thus, I would disagree with the authors that the intensively vegetated area (lichens) near the lake is a result of guano from a penguin colony and there are many other areas in the Antarctic where similar carpets of vegetation grow without the presence of guano substrate. In fact, so far there are no ornithogenic soils from Ardley Island that have been dated older than about 1000 years and none that occur within the catchment area of the lake as posited by the authors in their supplementary information (p. 10). The bio-elements from guano recovered from

the lake would thus represent very limited deposition from penguins molting near the lake or passing by it, not breeding next to or within the catchment area of the lake. Skuas may also account for some of these bio-elements as they currently nest all around the lake and use it for bathing, and have probably done so as long as the lake has been there. Could their activities also be responsible for some or most of the bio-elements found in the lake sediments? Otherwise, why do the lake sediments date much older than the terrestrial record of the penguin occupation? A similar phenomenon occurs at Lake Boeckella at Hope Bay where Zale (1994) dated sediments from a core to the mid-Holocene, but there are no ornithogenic soils or abandoned penguin colonies around this lake older than about 600 years. While it is possible that volcanic eruptions and ash deposits may have buried older ornithogenic soils (e.g., mid Holocene), the shallow bedrock at both Ardley and Hope Bay do not have the deposition or any deeply buried ornithogenic soils that correspond in age with the lake deposits at those locations. So, there is a mismatch in data from lakes versus existing ornithogenic soils, at least at these two sites that is very puzzling. While it is possible that volcanic ash deposits may have buried older ornithogenic soils at some locations (for example in the Chinstrap colony at Bailey Head, Deception Island, where exposed deposits show layers of volcanic ash above and below penguin remains) this does not seem to be the case at both Ardley Island and Hope Bay.

Last, I find it difficult to believe that a volcanic eruption would cause abandonment of penguin colonies for hundreds of years (Line 221). Penguins are quick to recover and recolonize sites that are ice-free, as long as sufficient food is available nearby, and this seems to be the case as observed in the layering at Bailey Head. If the volcanic eruptions had caused climate change, and thus abandonment over centuries, then the climate data should have corresponded better with the occupation history for the penguins. I do believe the authors are on to something here with these data, and the negative correlation of penguin occupation with the tephra layers is pretty strong, but unless the volcanic eruptions were continuous over hundreds of years, how could single or a few eruptions at one time cause abandonment for such long periods? The dates on Gentoo penguin remains from Ardley Island also include periods when the island was supposedly abandoned, so there is another mismatch there as well. What is the source for the statement on new colonies being established in the southern to mid Antarctic Peninsula as others are abandoned (last sentence on p. 10, Lines 227-230)? The only relatively new Gentoo colonies I know of in this part of the peninsula (e.g., Palmer Station) are the result of expansion of Gentoos to the south in relation to climate warming.

Figure 5 (d) refers to ornithogenic soils in the lake record, but technically these are not soils but ornithogenic sediments. Ornithogenic soils have a specific lithology and formation at penguin colonies, but the lake sediments are only influenced by penguin guano draining from areas around the lake and are thus usually distinguished from the soils as ornithogenic sediments.

(e) What does SSI, NAP, and S-Mid AP stand for? I'm guessing South Shetland Islands, Northern Antarctic Peninsula, and southern to middle AP, but these acronyms need to be defined in the figure caption or in the text where Figure 5 is cited. Also, how do you distinguish Adelie penguin guano, or even *Pygoscelis* guano, from that of other species as labeled in this part of the figure? Do the penguin dates shown by the solid green squares include the bones from their sediment core at Ardley Lake? If so, were they corrected for the marine carbon reservoir effect? How do the other penguin dates from Ardley Island (published by Emslie et al. 2013) fit in here? More clarification in this part of the figure is needed.

Overall this study is an excellent contribution to the effects of volcanism and penguin occupation in the Antarctic Peninsula. Once missing information is included and addressed, as well as questions raised above, it will be a much more thorough review of the topic and of broad appeal to readers of Nature Communications.

Reviewer #2 (Remarks to the Author):

Review of NCOMMS-16-06888 Morien et al: Past penguin population fluctuations linked to catastrophic volcanism.

Tom Hart, Department of Zoology, Oxford University (signed review, I am happy to be contacted directly if the authors require clarification or disagree with any statement).
tom.hart@zoo.ox.ac.uk

Limitations of review: I am a population ecologist studying population changes in penguins over time. I no experience of stable isotopes other than through collaborators, so I have explicitly not commenting on the validity of these methods; I assume at least one reviewer is an expert in the method.

Summary: I think this is a very interesting manuscript and I think it is suitable for publication in Nature Communications, but with one important caveat; that they recognise that they are looking largely at local effects more than regional or general trends - I think the strengths and weaknesses of the inference need to be more explicit. Regional long-term demographic trends have already been inferred by genetic methods and similar stable isotopic studies on occupancy have been carried out in the Antarctic Peninsula. The authors provide a much finer scale resolution of occupancy than previously published and if carried out over more sites would open up long-term ecology of penguins in space and time. In particular, I think it's applicable for Nature Communications because there have been a number of high - impact studies on climate change in Antarctica and its impact on penguins and other predators. None that I'm aware of so far have been able to attribute population changes to volcanism and therefore some impacts that have been attributed to climate change and sea ice fluctuations may need to be reinterpreted. In particular, past biological responses to environmental change and natural phenomena are of wider interest, which is why I think it is ultimately suitable for Nature Communications. I've often wondered what volcanism might mean for penguin populations in the Scotia Arc, as (for example) Zavodovski Island holds a quarter of the world's chinstrap penguins and yet must periodically become uninhabitable. I'm delighted that the authors have been able to investigate this, although I have some caveats about interpretation and I query some of the impacts being direct.

Finally, I'm not sure that the authors sufficiently recognise the different responses between the pygoscelid species, which are vital to the interpretation of these data. I would encourage the authors to rewrite their introduction and discussion with more caveats so that it is clear what they are doing that is new in regards to the literature, while reflecting

Specific points:

Line 58: "Our knowledge of penguin species response to climate and environmental changes is, however, largely based on the short term observational record (typically less than 60 years) and archaeological studies of abandoned penguin rookeries".

The claim that this is the first study able to infer populations over long term is incorrect. In particular, in the last two years, there have been several studies addressing exactly this using the coalescent approach on genetic (Slack et al., 2006, Clucas et al., 2014, Younger et al., 2015) or genomic (Trucchi et al., 2014) data with 1000-10,000 year resolution (Li et al., 2014). However, there is spatial and temporal averaging in such studies, so they miss the fine scale dynamics that this technique captures. Most importantly, all previous studies have missed volcanism as a predictor of population change or movement.

L139 Zone is confusing- can this be something with less spatial connotations?

L152 "our Ardley Lake record provides the oldest evidence of penguin occupation on the Antarctic Peninsula". Again, not really as Clucas et al (Clucas et al., 2014) showed a southern population and Levy (Levy et al., 2016) showed that this is distinct to the peninsula region in gentoos. As

these are inferred by genetics, you've provided the first direct and fine scale occupancy, which is excellent. Maybe just change to "oldest direct evidence at a specific site" or words to that effect?

L219-230 This whole paragraph seems out of place in the South Shetland Islands - in the South Sandwich Islands, colonies are directly affected by volcanism; they nest on volcanic slopes on most of the islands. Volcanism melts snow and there is evidence of deep tephra deposits indicating obliteration of colonies, with is evidence of mass mortality. However, this seems a stretch for Ardley Island - Deception is 130 km away from Ardley and that would have to be a cataclysmic event for ash to cover the colony. So, can the authors clarify whether ash is actually covering the breeding sites? If not, it would seem that the Deception eruptions have an indirect effect. Can they think of any indirect effects, or might Deception coincide with local landslides etc? Personally, I am sceptical that any direct impact of an eruption would be as dramatic as this on a distant colony.

L230 Fig 5e is unclear and doesn't provide much evidence to support the idea that penguins have moved South during this period. There is reasonable evidence for an inverse relationship between F.o.s at Adley Lake and tephra, but the relationship between tephra and penguin occupation records S-Mid AP are not conclusive. I can see as many incidences of occupation where there is tephra as without. Moreover, where occupancy cannot be attributed to one species or to a species other than Gentoo, there is a really important distinction to be made. As the authors state in the introduction, Gentoo penguins are near shore, generalist foragers that are ice-intolerant compared to the more krill obligate, more ice tolerant chinstrap and Adelies. We know that these species do not respond the same way in terms of populations or breeding activity (Lynch et al., 2012a, Lynch et al., 2012b).

Why do these colonies take so long to recover - what is the precision of the dating in this regard? I think this needed slightly more detailed consideration as it's a surprising result given the levels of mobility we see in Gentoo penguins.

References:

- CLUCAS, G. V., DUNN, M. J., DYKE, G., EMSLIE, S. D., LEVY, H., NAVEEN, R., POLITO, M. J., PYBUS, O. G., ROGERS, A. D. & HART, T. 2014. A reversal of fortunes: climate change 'winners' and 'losers' in Antarctic Peninsula penguins. *Sci. Rep.*, 4.
- LEVY, H., CLUCAS, G. V., ROGERS, A. D., LEACHE, A. D., CIBOROWSKI, K. L., POLITO, M. J., LYNCH, H. J., DUNN, M. J. & HART, T. 2016. Population structure and phylogeography of the Gentoo Penguin (*Pygoscelis adeliae*) across the Scotia Arc. *Ecol. Evol.*
- LI, C., ZHANG, Y., LI, J. W., KONG, L. S., HU, H. F., PAN, H. L., XU, L. H., DENG, Y., LI, Q. Y., JIN, L. J., YU, H., CHEN, Y., LIU, B. H., YANG, L. F., LIU, S. P., ZHANG, Y., LANG, Y. S., XIA, J. Q., HE, W. M., SHI, Q., SUBRAMANIAN, S., MILLAR, C. D., MEADER, S., RANDS, C. M., FUJITA, M. K., GREENWOLD, M. J., CASTOE, T. A., POLLOCK, D., GU, W. J., NAM, K., ELLEGREN, H., HO, S. Y. W., BURT, D. W., PONTING, C. P., JARVIS, E. D., GILBERT, M. T. P., YANG, H. M., WANG, J., LAMBERT, D. M., WANG, J. & ZHANG, G. J. 2014. Two Antarctic penguin genomes reveal insights into their evolutionary history and molecular changes related to the Antarctic environment. *Gigascience*, 3.
- LYNCH, H. J., FAGAN, W. F., NAVEEN, R., TRIVELPIECE, S. G. & TRIVELPIECE, W. Z. 2012a. Differential advancement of breeding phenology in response to climate may alter staggered breeding among sympatric pygoscelid penguins. *Marine Ecology Progress Series*, 454, 135-4.
- LYNCH, H. J., NAVEEN, R., TRATHAN, P. N. & FAGAN, W. F. 2012b. Spatially integrated assessment reveals widespread changes in penguin populations on the Antarctic Peninsula. *Ecology*, 93, 1367-1377.
- SLACK, K. E., JONES, C. M., ANDO, T., HARRISON, G. L., FORDYCE, R. E., ARNASON, U. & PENNY, D. 2006. Early penguin fossils, plus mitochondrial genomes, calibrate avian evolution. *Molecular Biology and Evolution*, 23, 1144-1155.
- TRUCCHI, E., GRATTON, P., WHITTINGTON, J. D., CRISTOFARI, R., LE MAHO, Y., STENSETH, N. C. & LE BOHEC, C. 2014. King penguin demography since the last glaciation inferred from genome-wide data. *Proceedings of the Royal Society B: Biological Sciences*, 281.

YOUNGER, J. L., CLUCAS, G. V., KOOYMAN, G., WIENECKE, B., ROGERS, A. D., TRATHAN, P. N., HART, T. & MILLER, K. J. 2015. Too much of a good thing: sea ice extent may have forced emperor penguins into refugia during the last glacial maximum. *Global Change Biology*, n/a-n/a.

Reviewer #3 (Remarks to the Author):

Based on new data from the northern Antarctic Peninsula (N-AAP), Monien et al suggest a major role of volcanic eruptions on penguin populations over the Holocene and, somehow, a second role of climate. Although this finding is not valid for regions such as the Ross Sea, where volcanism has not been intense over the past 10,000 years, it represents a very interesting case study that will be attractive to a large audience. The manuscript is well written and well illustrated but I have several concerns both on the data/interpretations and the structure/focus of the manuscript that may prevent publication in its present form.

Climate conditions

Authors first claim that climate variations were not the main driver of penguin populations' dynamics over the Holocene. Holocene climate conditions in N-AAP are here mainly based on a new marine record offshore Anvers Island (NW-AAP) and JRI ice core (NE-AAP). Few other marine and lake records are cited but not shown in the main figures or supplementary figures, which make the reconstruction of regional climate uneasy. I therefore did it myself. I plotted the most recent, high-resolution ocean temperature and sea ice records from N-AAP on top of the climatic records presented in the manuscript. The bottom line is that the sea ice record presented in figure 5 does not agree with regional climate. Indeed, the authors suggest:

- lines 176-186: a warmer mid-Holocene between 6-3 ka BP based on lower *Fcurta/Fkerguelensis* ratio values and warmer air temperatures in JRI ice core. However, Heroy et al. (2008) in Bransfield Strait, Shevenell et al. (2011), Pike et al. (2011) and Etourneau et al. (2013) in Palmer Deep, Sterken et al. (2012) and Barbara et al (2016) in Prince Gustav Channel all show that cool and icy conditions prevailed during the 5-2 ka BP period.

- lines 192-193: similarly, the neoglacial (2.5-0.5 ka in the present study) is not as well expressed regionally as authors indicate. Shevenell 2011 and Etourneau 2013 showed a warming between during the 2-0.5 ka BP period while Pike 2012 inferred icy conditions during the 2-1.5 ka BP period and less icy conditions from 1.5 ka BP onward. In Bransfield Strait, Heroy 2008 suggested cooler conditions between 2.6-0 ka BP while Barcena et al. (1998) inferred warmer, less icy conditions between 1.5-0.5 ka BP and a subsequent cooling. Around JRI, Barbara 2016 showed warmer and less icy conditions during 2-0.5 ka BP and a subsequent cooling.

- line 184: Citing here Pike 2011 is wrong as their *Fcylindrus* record indicate cool and icy conditions between 6-1.5 ka BP except for an environmental amelioration around 2.7-2.2 ka BP. A warm MHH at 4.2-3.4 ka BP is right in their maximum sea ice event.

- line 188: Note that Yoon et al. (2000) presented their results in corrected ages, not calibrated. Calibration will shift their record, which may not be in phase with authors' own results.

JRI is the closest ice core covering the Holocene but its record is far from been understood.

Mulvaney et al. (2012) related JRI Holocene temperature trend to regional AAP climate evolution as it fits with Shevenell's oceanic temperature records until 2 ka BP. When the relationship subsequently shattered they invoked the Antarctic Dipole forced by the ENSO to explain climate differences between W-AAP and E-AAP. However, it is worth noting that millennial-scale oscillations due to the ENSO appeared at 5 ka BP (Moy et al., 2002; Conroy et al., 2008; Chazen et al., 2009), a time where JRI and W-AAP were still in phase. This casts doubt on the regional signal preserved in JRI, which probably depends on the origin of the precipitations as a result of the position of the Admunsen Sea Low. To note that JRI record is generally out-of-phase with the other marine records from N-AAP.

In conclusion, I would recommend that the Holocene regional climate is refined via the abundant literature. Other records should be plotted alongside the ones already presented for a direct

comparison, at least in the supplementary material.

I am also surprised that no other records from Ardley Lake and Ardley Island, nor other lake records from the South Shetland Islands, are presented for a better characterization of regional "continental" climate.

Structure / focus of the manuscript

If the lack of relationship between penguin populations and the new, more exhaustive regional climate reconstruction still persists, I would recommend that authors go further on the link with volcanism. The abruptness of colony disappearance does argue for a rapid trigger that is probably not climatic. Volcanism therefore seems a good explanation, as supported by the data. However, I would like to see more discussion on the relationships between volcanism and colony dynamics. Is colony disappearance linked to eruption intensity, duration, repetition? Why is re-colonization timing very different (quick in Phase 2, slow in Phase 4)? How come Phase 3 occurred during a strong volcanic activity period? Do populations die on site or migrate during an eruption? Volcanisms should be introduced in the introduction that is currently, and somehow confusingly, climate oriented. All this will make a rather descriptive paper much more interesting to a larger audience.

Additional comments

It is not clear to me where penguins' colonies were located on Ardley Islands in the past. Today, they seem located on its eastern side, while Ardley Lake stands on its western side. The lake seems presently separated from the active colonies by "mountains". As there is no information on the draining basin of Ardley Lake it is difficult to know whether the location of active colonies can bias the ornithogenic records. I mean no signal in the lake despite the presence of an important colony because the ornithological soil are not transported in the lake.

Line 224: Authors argue that volcanic eruptions induced "high local mortality". I did not get whether they are macro-evidences (more bones) to support this. Can't we imagine that penguins just moved to another nesting site?

Figure 2 demonstrates two evident clusters in Phase 4. The more ancient cluster appears related to clusters found during CONISS zones 6 and 8 when low REE are encountered. Is Phase 4 "a" really a high occupation period?

Figure 5 shows out-of-phase tephras in Ardley Lake and nearby Yanou Lake. Pretty weird. Why? Dating uncertainties? Phase 3 is not represented by a white box in *P. sp guano* (Fos >10%). MCA marked during cool air temperatures and high sea ice presence. Weird.

Supplementary material

Chronology

It is noted that freshwater samples were calibrated with OXCAL why marine samples were calibrated using marine13. Was it in OXCAL too? It is then stated that the age model was developed with BACON. Were the calibrated ages injected into BACON? If so, why not using BACON to calibrate the raw 14C ages? If not, why first using OXCAL?

It is strange that the total reservoir age correction is 1870 years from the Anvers Shelf record (quite offshore location) and 1064 years in the marine part of the lake core (probably semi-enclosed basin). With old ice runoffs and slow ocean circulation south of South Shetland Islands I would have expected the opposite. Let's suggest that the AR correction is too small and was around 1900 years as in W-AAP core, what will it change to the interpretations vs climate?

Sea ice record

Fcurta and *Fkerguelensis* records are generally anti-correlated in Southern Ocean records. What is here the point to present their ratio that may only increase the amplitude of the Holocene variations? I would like to see their raw data along with other species. *Fcylindrus*, *Tantarctica1* and *Tantarctica2* for example for a better assessment of local environmental conditions.

Other

Phase 4 is noted between 111.9-67.6cm in Supp. Table 5 while between 155-67.6 cm in Figure 2. This confirms my previous comment that the Gentoo colony was probably not that large during

Phase4"a".

Our response to specific points raised are shown in **bold** below, and we have revised the main text and Supplementary Information files accordingly.

Reviewers' comments:

Reviewer #1 (Remarks to the Author):

This paper provides an extensive analysis of lake sediments from a core collected on Ardley Island, King George Island, Antarctic Peninsula. The authors identify sequences of bio-elements in the core that are characteristic of penguin guano to infer occupation of the island by breeding penguins during different time periods over the past 8,000 yrs. They further posit that volcanic activities have curtailed or prevented occupation of this area during periods when no bio-elements are present in the sediments. Instead, a negative correlation between tephra deposits and the bio-elements suggests that volcanic eruptions have caused abandonment of the island over millennia. Their data also provide the oldest evidence for penguin occupation in the northern Antarctic Peninsula. The inclusion of a second core from a control lake (Yanou Lake), situated far from any penguin influence, is an excellent addition to the study and quite useful in verifying the bio-element input from guano at Ardley Island as well as the negative correlation with the tephra. The geochemical and statistical analyses are sound and appropriate and the resulting patterns they found in relation to volcanic events are compelling and novel. There are a few places where the paper could use more clarification and detail, and where the authors could address discrepancies in the data from their and other studies at Ardley Island and the northern Antarctic Peninsula.

There are some key citations that were missed and a few corrections that are needed. First, the authors refer to ancient penguin remains throughout the paper as 'archaeological' but this term refers strictly to human artefacts and remains. While archaeological methods have been used to excavate ancient penguin sites, the remains here should be referred to as 'sub-fossil' and not archaeological.

On line 50, the scientific name for Adélie penguin does not have the accent mark above the "e".

They should also avoid using the term 'rookery' (Line 60) as that is an old term for a breeding colony that originates from rook colonies in Europe. They should instead consistently refer to penguin breeding sites as colonies.

We have replaced 'archaeological' with 'sub-fossil' throughout the text and have updated citations to include all key relevant references. We have replaced 'rookery' with 'colony' throughout the text. All occurrences of Adélie now have an accented "e".

In addition, the Chinese research on Ardley Island is well cited by the authors, but they have published at least two other papers (Sun et al. 2001, Chinese Journal of Polar Science and Wang et al. 2007, Polar Biology) on bio-elements from a sediment core from Ardley Island (which they refer to as sample Y2 from Antarctic West Lake, just west of Ardley Lake). The authors have already compared their data to other published data from this Y2 core (their citations 18, 20, and 41), but should also look at these papers, especially the more recent one, and the conclusions drawn by the authors on the Gentoo occupation.

In addition, another study that reports mid-Holocene penguin remains from King George Island is by del Valle et al. (2002, Antarctic Science) and a more recent study on the age of the current Gentoo penguin colony on Ardley Island, which extends to 800-1000 BP, is in Emslie et al. (2013, Antarctic Science).

We now reference the Sun/Wang et al papers. These and other similar papers do not contain sufficient geochemical and chronological data to enable us to include them in our analysis and comparison plots. We requested a copy of the Sun et al.³ dataset from the authors, but received no response, despite several requests. Although their sampling resolution is low, we had noticed a similar pattern of bio-element change (e.g., S, P₂O₅, Cu, Zn, Se, Sr) in our record to that in the Sun et al.² 75–25 cm depth profile. We assume that the Wang et al.¹ chronology is based on the same age-depth model as the Sun et al.² paper. Assuming that the published data have age-depth models that are comparable, the guano peak between 1,800-1,400 years

ago broadly coincides with the latter stages of an elevated guano peak in our record, but their main peak is offset by c. 400–500 years (towards present day). Differences between our data and the Sun et al.² data could reflect different radiocarbon calibration curves used, since the Sun et al.² data were most likely calibrated using the INTCAL98 Northern Hemisphere calibration curve. Unfortunately, we have not been able to verify this or update the published Lake Y2 data to the standardised SH13 calibrated chronology used in our paper. We have now updated all previous MARINE98 and 09 calibrations to MARINE13 using the local marine reservoir correction of $1,100 \pm 50$ ($\Delta R=700 \pm 50$) recommended by Emslie et al.⁶ (see also Supplementary Note 5). Calibration issues aside, in the discussion we now suggest that these shifts might reflect penguin colony migration away from the centre of the island towards its newly exposed periphery of Lake Y2 (summarised in Fig. 6). We note that the period of low temperature referred to by Sun et al. between 2,300–1,800 a BP refers to a paper by Clapperton et al.⁴ and is based on records from South Georgia. For the Antarctic Peninsula, this coincides with the start of a ‘neoglacial’ phase of late Holocene cooling with glacier readvances across the northern Peninsula region. We have also added data from del Valle et al.⁵ and Emslie et al.⁶ to the revised Supplementary Datafile and now summarise all sub-fossil data plot shown in Figure 5.

Finally, the authors indicate that three juvenile penguin bones were radiocarbon dated from their lake core (supplementary Fig. 4), but these dates are not included in supplementary Table 2 or reported anywhere else in the manuscript that I can find. Since these are the only direct dates on penguin remains from their lake core, and thus the penguin occupation, it is frustrating to not know the dates obtained. These bone dates also could help verify the age of the sediment core based on the other dates provided in Table 2.

In the revised version, we include recently acquired bone-collagen radiocarbon ages (Supplementary Table 2). As reviewer 1 suggests, we used this new data to verify age-depth models and discuss the differences between the original (ARD-M1) and new age-depth model (ARD-M5) in detail in Supplementary Note 7. In summary, the new ARD-M5 age depth model in the revised version is well-constrained and consistent, but has a younger, 8–3.5 ka cal BP age-depth profile compared to the ARD-M1 age depth model (see below and Supplementary Notes 5–7 and Supplementary Figs. 4, 5 for details).

Figure 1. Above left: Comparison of original Ardley Lake record BACON age depth model ARD-M1 with revised ARD-M5 model that includes bone-collagen ages and marine reservoir correction for bulk sediment ages from guano-phases. The latter was applied because the %C record is strongly and significantly correlated with the %F_{o.s.ed.}, indicating that carbon in the guano-phases likely has a large penguin-derived marine component. Above right-top: comparison between the %F_{o.s.ed.} profile for ARD-M1 and ARD-M5

highlighting the mid-late Holocene ‘younger-shift’. Above right-bottom: plot comparing ARD-M5 and YAN-M4 age-depth model sedimentation rates, highlighting good concordance between the c. 6–4.5 cal ka BP phase of elevated sedimentation rates in both cores. The main tephra layer deposition ages are also better aligned when using the ARD-M5 age-depth model.

Overall, a more comprehensive summary of the new data presented by the authors with these additional data will produce a well-rounded and excellent contribution to the literature on penguin occupations in the northern Antarctic Peninsula.

Another topic worthy of additional discussion is that, while Ardley Lake is not too far from an active Gentoo penguin colony on the east end of the island, the area around the lake itself has shallow rocky soil, shallow bedrock, and no evidence for abandoned penguin colonies anywhere near the lake. Thus, I would disagree with the authors that the intensively vegetated area (lichens) near the lake is a result of guano from a penguin colony and there are many other areas in the Antarctic where similar carpets of vegetation grow without the presence of guano substrate.

We agree that there are many areas with a similarly dense vegetation on the Peninsula and that NDVI doesn’t always reflect guano fertilisation (see Casanovas et al.⁷) and have expanded Supplementary Note 3 to discuss the NDVI analysis in more detail. It is interesting to note that Casanovas et al.⁷ found that: ‘Moss richness was correlated with isolation and latitude at the local scale, while lichen richness was correlated with summer mean temperature and, for 17 sites where penguins were present with <20,000 breeding pairs, penguin colony size’, implying that penguin colonies have a fertilising effect, but only up to a certain point, after which, they become destructive.

In fact, so far there are no ornithogenic soils from Ardley Island that have been dated older than about 1000 years and none that occur within the catchment area of the lake as posited by the authors in their supplementary information (p. 10). The bio-elements from guano recovered from the lake would thus represent very limited deposition from penguins moulting near the lake or passing by it, not breeding next to or within the catchment area of the lake. Skuas may also account for some of these bio-elements as they currently nest all around the lake and use it for bathing, and have probably done so as long as the lake has been there. Could their activities also be responsible for some or most of the bio-elements found in the lake sediments? Otherwise, why do the lake sediments date much older than the terrestrial record of the penguin occupation? A similar phenomenon occurs at Lake Boeckella at Hope Bay where Zale (1994) dated sediments from a core to the mid-Holocene, but there are no ornithogenic soils or abandoned penguin colonies around this lake older than about 600 years. While it is possible that volcanic eruptions and ash deposits may have buried older ornithogenic soils (e.g., mid Holocene), the shallow bedrock at both Ardley and Hope Bay do not have the deposition or any deeply buried ornithogenic soils that correspond in age with the lake deposits at those locations. So, there is a mismatch in data from lakes versus existing ornithogenic soils, at least at these two sites that is very puzzling. While it is possible that volcanic ash deposits may have buried older ornithogenic soils at some locations (for example in the Chinstrap colony at Bailey Head, Deception Island, where exposed deposits show layers of volcanic ash above and below penguin remains) this does not seem to be the case at both Ardley Island and Hope Bay.

We also did not find ‘ornithogenic soil’ sequences around Ardley Lake, or elsewhere on Ardley Island. We discuss this point in the revised Introduction and Supplementary Note 2, where we relate it to highly active erosion on Ardley Island and note that lake sediments protected by a permanent water column provide a more continuous and undisturbed record of past events. New sub-fossil radiocarbon bone-collagen ages from juvenile (Gentoo) penguins in the ARD record show that penguins laid eggs on the island from at least 5,040 cal a BP [5,240–4,860 cal a BP] (Supplementary Fig. 4, Supplementary Table 2). Ardley Lake could have been a moulting site in the past, as it is not possible to distinguish from our data exactly how penguins used the Ardley Lake catchment. We now reconstruct fluctuations in the numbers of penguins potentially present in the catchment in a new penguin population modelling section. Figure 5 summarises results and the revised Methods section contains full details of how these values were calculated. A key factor to bear in mind is that, during the mid Holocene, the amount of land available to penguin colonies on Ardley Island was largely restricted to the central area around Ardley Lake (Fig. 6). The presence of penguin bones at

several points within the Ardley Lake core and the abundance, longevity and repetitious nature of the guano-enhanced bio-element signal in our lake sediment core argue against the 'fleeting visit scenario' suggested by Reviewer 1 and point towards phases of more permanent occupation. Skuas nest around Ardley Lake today, but there is no significant biogeochemical signal in the most recent surface sediments in Ardley or Yanou lakes (where a similar density of skua nests exists). We observed very few skua nests in the Ardley lake catchment when coring – probably, at most, about four and census data for King George Island shows that the number and density of skua nests is orders of magnitude less than penguins (see www.penguinmap.com/mapppd for data). We consider that our guano-phase definition based on the *F_{o, sed.}* weighted mean threshold at 10% excludes the possibility of deposition by only a small number of passing penguins or skuas nesting in the catchment.

Last, I find it difficult to believe that a volcanic eruption would cause abandonment of penguin colonies for hundreds of years (Line 221). Penguins are quick to recover and recolonize sites that are ice-free, as long as sufficient food is available nearby, and this seems to be the case as observed in the layering at Bailey Head. If the volcanic eruptions had caused climate change, and thus abandonment over centuries, then the climate data should have corresponded better with the occupation history for the penguins. I do believe the authors are on to something here with these data, and the negative correlation of penguin occupation with the tephra layers is pretty strong, but unless the volcanic eruptions were continuous over hundreds of years, how could single or a few eruptions at one time cause abandonment for such long periods?

We address these points in the revised Discussion section, in Supplementary Note 9 and in the Supplementary Discussion section. We have also added new Tables 1 and 2 to the main paper to summarise the length of guano-phases and main recovery intervals. We agree that volcanic impacts might not last for several hundred years following a large eruption. Rather, our recovery periods reflect the amount of time that it takes the colony to rebuild to a point where their guano constitutes >10% of the sedimentary input into the lake. Unlike some other species, gentoos do not have a strong collective memory of former colony location, especially if a previously occupied site was completely abandoned. We note that the longest recovery interval occurred during a series of mid Holocene T5 eruptions during a period of increased climate instability on the northern Peninsula. We do not link this period of climate instability to volcanic activity since eruptions from Deception Island are probably not large enough to influence regional climate for a prolonged period of time. Rather this combination of these events were particularly unfavourable for more rapid post-eruption recolonization.

The dates on Gentoo penguin remains from Ardley Island also include periods when the island was supposedly abandoned, so there is another mismatch there as well.

In the revised version, the inclusion of bone-collagen age data and other improvements to the ARD age-depth model (ARD-M5) produced a more coherent relationship between eruption ages in the ARD and YAN records (see Figs. 4, 5, Tables 1, 2). The larger eruptions now correspond to periods when penguin bones have not been found, for example 5–4 ka and 2.7–3.5 ka. Gaps in the Peninsula sub-fossil record also correspond with the absence of radiocarbon dated penguin bones and/or low *F_{o, sed.}* values in the Ardley Lake sediment core.

What is the source for the statement on new colonies being established in the southern to mid Antarctic Peninsula as others are abandoned (last sentence on p. 10, Lines 227-230)? The only relatively new Gentoo colonies I know of in this part of the peninsula (e.g., Palmer Station) are the result of expansion of Gentoos to the south in relation to climate warming.

We have revised the Discussion to address this point. One possibility is that the delayed response evident in the Peninsula sub-fossil occupation record shortly after the Ardley Lake T5 and T4 post-eruption colony decline relates to new colonies becoming established in the mid-southern AP c. 5.4 and 4.2 cal ka BP (Fig. 5c, d).'

Figure 5 (d) refers to ornithogenic soils in the lake record, but technically these are not soils but ornithogenic sediments. Ornithogenic soils have a specific lithology and formation at penguin colonies, but the lake

sediments are only influenced by penguin guano draining from areas around the lake and are thus usually distinguished from the soils as ornithogenic sediments. (e) What does SSI, NAP, and S-Mid AP stand for? I'm guessing South Shetland Islands, Northern Antarctic Peninsula, and southern to middle AP, but these acronyms need to be defined in the figure caption or in the text where Figure 5 is cited. Also, how do you distinguish Adelie penguin guano, or even *Pygoscelis* guano, from that of other species as labelled in this part of the figure? Do the penguin dates shown by the solid green squares include the bones from their sediment core at Ardley Lake? If so, were they corrected for the marine carbon reservoir effect? How do the other penguin dates from Ardley Island (published by Emslie et al. 2013) fit in here? More clarification in this part of the figure is needed.

We now refer to $F_{o, sed.}$ rather than $F_{o, s.}$ and have clarified all locational definitions. It is not possible to distinguish between species on the basis of guano geochemical signal, but juvenile Gentoo penguin bones (identified at the Natural History Museum) were found in combination with the guano peaks and in combination with diatom evidence for the existence of shallow pools of guano-influenced water around the lake. The species identification used in the original diagram was based on the as-published sub-fossil evidence, not our data. As described above, we added Emslie et al. (2013)⁶ data and recalibrated all published data using MARINE13 curves. We removed the species distinction part of Figure 4 as it was confusing and as we did not have sufficient species information for it to be meaningful. We include new bone-collagen ages in the revised version of Figure 4 and distinguish these from bones that have age-depth modelled ages (see Figure 5 Legend). We included a marine reservoir effect as detailed in Supplementary Note 5.

Overall this study is an excellent contribution to the effects of volcanism and penguin occupation in the Antarctic Peninsula. Once missing information is included and addressed, as well as questions raised above, it will be a much more thorough review of the topic and of broad appeal to readers of Nature Communications.

Reviewer #2 (Remarks to the Author):

Review of NCOMMS-16-06888 Monien et al: Past penguin population fluctuations linked to catastrophic volcanism.

Tom Hart, Department of Zoology, Oxford University (signed review, I am happy to be contacted directly if the authors require clarification or disagree with any statement). tom.hart@zoo.ox.ac.uk

Limitations of review: I am a population ecologist studying population changes in penguins over time. I no experience of stable isotopes other than through collaborators, so I have explicitly not commenting on the validity of these methods; I assume at least one reviewer is an expert in the method.

Summary: I think this is a very interesting manuscript and I think it is suitable for publication in Nature Communications, but with one important caveat; that they recognise that they are looking largely at local effects more than regional or general trends - I think the strengths and weaknesses of the inference need to be more explicit. Regional long-term demographic trends have already been inferred by genetic methods and similar stable isotopic studies on occupancy have been carried out in the Antarctic Peninsula. The authors provide a much finer scale resolution of occupancy than previously published and if carried out over more sites would open up long-term ecology of penguins in space and time. In particular, I think it's applicable for Nature Communications because there have been a number of high - impact studies on climate change in Antarctica and its impact on penguins and other predators. None that I'm aware of so far have been able to attribute population changes to volcanism and therefore some impacts that have been attributed to climate change and sea ice fluctuations may need to be reinterpreted.

In particular, past biological responses to environmental change and natural phenomena are of wider interest, which is why I think it is ultimately suitable for Nature Communications.

I've often wondered what volcanism might mean for penguin populations in the Scotia Arc, as (for example) Zavodovski Island holds a quarter of the world's chinstrap penguins and yet must periodically become

uninhabitable. I'm delighted that the authors have been able to investigate this, although I have some caveats about interpretation and I query some of the impacts being direct.

We address the general caveats raised above revising the introductory paragraph and including a new section in the Discussion, Supplementary Note 9 and a new section in the Supplementary Discussion. New diagrams illustrate the potential scale and impact of Deception Island eruptions. For example, in the Discussion and Supplementary Discussion we address comments relating to the impact of volcanic eruptions and penguin mortality.

Finally, I'm not sure that the authors sufficiently recognise the different responses between the pygoscelid species, which are vital to the interpretation of these data.

I would encourage the authors to rewrite their introduction and discussion with more caveats so that it is clear what they are doing that is new in regards to the literature, while reflecting

This comment was incomplete in the version we were sent, finishing mid-sentence. We recognise and understand different species responses and have clarified this section in revised introduction. We consulted Prof. Phil Trathan (BAS) and Dr. Claire Waluda, experts in seabird ecology, and they have read the original paper and revised version. The revised version contains additional novel aspects and more clearly sign-posts what is new. In summary, this study is the first to: 1) Establish a long-term link between volcanism and penguin colony change; 2) Present reconstruct quantitative terrestrial temperatures from lake sediments on the Antarctic Peninsula (the Yanou Lake GDGT record); 3) Undertake quantitative penguin population modelling using biogeochemical data from lake sediments on the AP; 4) Include a control site to test the biogeochemical signal of penguin absence and use the same control site to reconstruct a climate record which is most relevant to the guano record; 5) Statistically test the relationship between potential drivers of penguin colony change using quantitative palaeoclimate data.

Specific points:

Line 58: "Our knowledge of penguin species response to climate and environmental changes is, however, largely based on the short term observational record (typically less than 60 years) and archaeological studies of abandoned penguin rookeries".

The claim that this is the first study able to infer populations over long term is incorrect. In particular, in the last two years, there have been several studies addressing exactly this using the coalescent approach on genetic (Slack et al., 2006, Clucas et al., 2014, Younger et al., 2015) or genomic (Trucchi et al., 2014) data with 1000-10,000 year resolution (Li et al., 2014). However, there is spatial and temporal averaging in such studies, so they miss the fine scale dynamics that this technique captures. Most importantly, all previous studies have missed volcanism as a predictor of population change or movement.

Here, we were referring more specifically to in-situ studies that involve radiocarbon dating of bones or sediments. We have noted developments in genetic studies in the revised version of the main text.

L139 Zone is confusing- can this be something with less spatial connotations?

This is a standard term widely used in palaeolimnology. We now refer to guano phases rather than zones to avoid confusion.

L152 "our Ardley Lake record provides the oldest evidence of penguin occupation on the Antarctic Peninsula". Again, not really as Clucas et al (Clucas et al., 2014) showed a southern population and Levy (Levy et al., 2016) showed that this is distinct to the peninsula region in gentoos. As these are inferred by genetics, you've provided the first direct and fine scale occupancy, which is excellent. Maybe just change to "oldest direct evidence at a specific site" or words to that effect?

We have updated the main text as suggested

L219-230 This whole paragraph seems out of place in the South Shetland Islands - in the South Sandwich Islands, colonies are directly affected by volcanism; they nest on volcanic slopes on most of the islands.

Volcanism melts snow and there is evidence of deep tephra deposits indicating obliteration of colonies, with evidence of mass mortality. However, this seems a stretch for Ardley Island - Deception is 130 km away from Ardley and that would have to be a cataclysmic event for ash to cover the colony. So, can the authors clarify whether ash is actually covering the breeding sites? If not, it would seem that the Deception eruptions have an indirect effect. Can they think of any indirect effects, or might Deception coincide with local landslides etc.? Personally, I am sceptical that any direct impact of an eruption would be as dramatic as this on a distant colony.

We address this point in sections of the new Discussion, Table 2, Supplementary Note 9 and in the Supplementary Discussion that deal with the potential causes of eruption-related penguin mortality.

L230 Fig 5e is unclear and doesn't provide much evidence to support the idea that penguins have moved South during this period. There is reasonable evidence for an inverse relationship between F.o.s at Adley Lake and tephra, but the relationship between tephra and penguin occupation records S-Mid AP are not conclusive. I can see as many incidences of occupation where there is tephra as without. Moreover, where occupancy cannot be attributed to one species or to a species other than Gentoo, there is a really important distinction to be made. As the authors state in the introduction, Gentoo penguins are near shore, generalist foragers that are ice-intolerant compared to the more krill obligate, more ice tolerant chinstrap and Adelies. We know that these species do not respond the same way in terms of populations or breeding activity (Lynch et al., 2012a, Lynch et al., 2012b).

We removed the species distinction from the diagram as was confusing and the published evidence did not have sufficient information to be meaningful. We have address other points in our summary response and in our response to Reviewer 1. In summary, eruption event age and guano mismatches are largely dealt with by the inclusion of a new age-depth model for the Ardley Lake sediments (ARD-M5) which includes bone-collagen ages and applies a marine reservoir correction to bulks sediment ages from guano phases - see Supplementary Note 7 for details.

Why do these colonies take so long to recover - what is the precision of the dating in this regard? I think this needed slightly more detailed consideration as it's a surprising result given the levels of mobility we see in Gentoo penguins.

We have added two new tables (Table 1 and 2), and revised the Discussion which show dating precision. However, it is important to note here that the main tephra deposits are stratigraphically-distinct, obeying the law of superposition in marking the end of guano phases GP-2, GP-3, GP-4 in the Ardley Lake record (Table 2).

References:

We have added most of the suggested references to the new introduction

Reviewer #3 (Remarks to the Author):

Based on new data from the northern Antarctic Peninsula (N-AAP), Monien et al suggest a major role of volcanic eruptions on penguin populations over the Holocene and, somehow, a second role of climate. Although this finding is not valid for regions such as the Ross Sea, where volcanism has not been intense over the past 10,000 years, it represents a very interesting case study that will be attractive to a large audience. The manuscript is well written and well-illustrated but I have several concerns both on the data/interpretations and the structure/focus of the manuscript that may prevent publication in its present form.

Climate conditions

Authors first claim that climate variations were not the main driver of penguin populations' dynamics over the Holocene.

Holocene climate conditions in N-AAP are here mainly based on a new marine record offshore Anvers Island (NW-AAP) and JRI ice core (NE-AAP). Few other marine and lake records are cited but not shown in the main figures or supplementary figures, which make the reconstruction of regional climate uneasy. I therefore did it myself. I plotted the most recent, high-resolution ocean temperature and sea ice records from N-AAP on top of the climatic records presented in the manuscript.

The bottom line is that the sea ice record presented in figure 5 does not agree with regional climate.

Indeed, the authors suggest:

- lines 176-186: a warmer mid-Holocene between 6-3 ka BP based on lower *Fcurta*/*Fkerguelensis* ratio values and warmer air temperatures in JRI ice core. However, Heroy et al. (2008) in Bransfield Strait, Shevenell et al. (2011), Pike et al. (2011) and Etourneau et al. (2013) in Palmer Deep, Sterken et al. (2012) and Barbara et al. (2016) in Prince Gustav Channel all show that cool and icy conditions prevailed during the 5-2 ka BP period.

- lines 192-193: similarly, the neoglacial (2.5-0.5 ka in the present study) is not as well expressed regionally as authors indicate. Shevenell 2011 and Etourneau 2013 showed a warming between during the 2-0.5 ka BP period while Pike 2012 inferred icy conditions during the 2-1.5 ka BP period and less icy conditions from 1,5 ka BP onward. In Bransfield Strait, Heroy 2008 suggested cooler conditions between 2.6-0 ka BP while Barcena et al. (1998) inferred warmer, less icy conditions between 1.5-0.5 ka BP and a subsequent cooling. Around JRI, Barbara 2016 showed warmer and less icy conditions during 2-0.5 ka BP and a subsequent cooling.

To clarify, we stated, and still find, that the relationship between penguin population change and climate was inconsistent. This part of the paper was not intended to review AP Holocene climate and sea-ice change. Instead, we examined records that were most relevant to penguin colony occupation and feeding patterns (i.e., for gentoos up to c. 20–30 km from their nesting sites) on the South Shetland Islands and the AP. In the revised version, the most relevant climate record is the new Yanou Lake GDGT MSAT temperature record. To provide broader context of regional climate variability, we also revisited our compilations of Antarctic Peninsula climate history (made for the Bentley et al.⁹ and Sterken et al.¹⁰ papers and summarised for the South Shetland Islands in the recent Antarctic Peninsula PAIS ice sheet reconstruction papers Bentley et al.¹¹; Ó Cofaigh et al.¹²). We also examined climate and sea-ice records suggested, selecting the Etourneau et al.¹³ Palmer Deep 0–200 m GDGT sea surface temp record as the most relevant SST record and undertook new statistical analysis using the 6–0 ka occupation period for penguins on Ardley Island as our baseline. In agreement with some AP records, the Etourneau et al. (2013)¹³ record and our Yanou Lake GDGT temperature record both show a period of, c. 6–4 ka (mid Holocene) cooling. Changes in the Anvers Trough sea-ice record correspond to the key findings in a forthcoming AP-wide sea-ice compilation paper by Allen et al.²¹. We retain this sea-ice record in the revised version because it is located astride the modern day summer sea-ice limit that runs up through the Bransfield Strait (Fig. 1a). Other sea-ice records referred to by Reviewer 3 are from nearer-shore (inner fjord) or NE Peninsula localities and we consider these less relevant to changes in sea-ice conditions around the South Shetland Islands. In the revised version, we also include the Milliken et al. (2009)⁸ sea-ice record from Maxwell Bay, South Shetland Islands (Fig. 4) and have revised Fig 1a to show the position of the summer sea-ice limit in relation to location of the Anvers Trough. We also compare these records to other AP sea-ice records in the revised Discussion.

- line 184: Citing here Pike 2011 is wrong as their *Fcylindrus* record indicate cool and icy conditions between 6-1.5 ka BP except for an environmental amelioration around 2.7-2.2 ka BP. A warm MHH at 4.2-3.4 ka BP is right in their maximum sea ice event.

- line 188: Note that Yoon et al. (2000) presented their results in corrected ages, not calibrated. Calibration will shift their record, which may not be in phase with authors' own results.

We calibrated the Yoon et al.¹⁴ data with MARINE13 curve. According to Yoon et al.¹⁴ there was a colder phase with more sea-ice and advanced glaciers in Maxwell Bay between 7,320–5,190 cal a BP (or 7,290–5,150 cal a BP using $\Delta R=700 \pm 50$ in revised version). This corresponds to a period when F_{curta}/F_{kerg} ratio and Pelagic Open water Species % data in the Anvers Trough core show reductions in spring/summer open water.

JRI is the closest ice core covering the Holocene but its record is far from been understood. Mulvaney et al. (2012) related JRI Holocene temperature trend to regional AAP climate evolution as it fits with Shevenell's oceanic temperature records until 2 ka BP. When the relationship subsequently shattered they invoked the Antarctic Dipole forced by the ENSO to explain climate differences between W-AAP and E-AAP. However, it is worth noting that millennial-scale oscillations due to the ENSO appeared at 5 ka BP (Moy et al., 2002; Conroy et al., 2008; Chazen et al., 2009), a time where JRI and W-AAP were still in phase. This casts doubt on the regional signal preserved in JRI, which probably depends on the origin of the precipitations as a result of the position of the Admunsen Sea Low. To note that JRI record is generally out-of-phase with the other marine records from N-AAP.

In conclusion, I would recommend that the Holocene regional climate is refined via the abundant literature. Other records should be plotted alongside the ones already presented for a direct comparison, at least in the supplementary material.

I am also surprised that no other records from Ardley Lake and Ardley Island, nor other lake records from the South Shetland Islands, are presented for a better characterization of regional "continental" climate.

Overall, we broadly agree with Reviewer 3's assessment that spatial and temporal discrepancies exist in palaeoclimate and sea-ice records from across the AP, and have refined our assessment of regional climate in the Discussion accordingly. We more clearly define 'warmer' periods as being greater than the 6–0 ka (weighted) mean 95% confidence interval upper bound, using the YAN-GDGT terrestrial temperature record as the most relevant local baseline terrestrial temperature record for penguin occupation on Ardley Island. Our revised results show that the mid Holocene of records examined were 'cooler' than the Early Holocene Optimum (EHO). Interestingly, the Yanou Lake GDGT record is also characterised by 'cool' mid Holocene temperatures found in the Palmer Deep and other marine records, but not present in the JRI record. All records have a phase of late Mid Holocene Hypsithermal (MHH) warmth. The post 2.5 ka neogacial cooling evident in the JRI record is expressed in the Yanou Lake GDGT temperature record, but the MCA is not very well-defined in the Yanou record. Post MCA cooling into the so-called 'Little Ice Age' also occurs in the Yanou record and has been found in other SSI records (e.g., Haas et al.¹⁵; Monien et al.¹⁶), but is not evident in the JRI record. The Barbara et al. (2016)¹⁷ data from the Vega Drift area was not considered relevant. We have summarised evidence from other South Shetland Island palaeorecords in the revised Discussion for a better characterisation of Holocene changes in 'continental' climate. We only include quantitative temperature datasets in our analysis and plots and note that some commonly used lake sediment/terrestrial proxies for temperature change (e.g., productivity) can multiple causes.

Structure / focus of the manuscript

If the lack of relationship between penguin populations and the new, more exhaustive regional climate reconstruction still persists, I would recommend that authors go further on the link with volcanism.

The abruptness of colony disappearance does argue for a rapid trigger that is probably not climatic. Volcanism therefore seems a good explanation, as supported by the data. However, I would like to see more discussion on the relationships between volcanism and colony dynamics. Is colony disappearance linked to eruption intensity, duration, repetition? Why is re-colonization timing very different (quick in Phase 2, slow in Phase 4)?

We have improved the volcanism aspect of this paper as recommended and as outlined in our overview and in our response to Reviewer 1. The timing of tephra deposition in Ardley and Yanou Lakes is better aligned using the new age-depth model for Ardley Lake.

How come Phase 3 occurred during a strong volcanic activity period?

The most disruptive part of the record is during the T5 eruption series. Full recovery into guano phase 3 (i.e., back to >10% $F_{o, sed.}$) only occurs at the end of this period of disruption (see revised Figures 2 and 4). We have now added the ash layers to the stratigraphic diagram (Figure 3) to more clearly show that phase 3 is bracketed by tephra deposits.

Do populations die on site or migrate during an eruption?

The answer to this would require similar reconstructions of *in-situ* penguin colony population change from several key locations across the Peninsula. Genetic evidence suggests southwards migration occurred at some point during the Lateglacial/Holocene, but genetic 'dating' uncertainties are probably too large to identify responses to specific eruptions or migration patterns following an eruption event.

Volcanisms should be introduced in the introduction that is currently, and somehow confusingly, climate oriented. All this will make a rather descriptive paper much more interesting to a larger audience.

We have added a new paragraph to the introduction and extended the last two paragraphs in the Discussion as recommended. Overall, we have made the paper more volcanism-orientated by clarifying the different scales of eruptions (see Figures 4 and 5) and including new datasets that better describe volcanic aspects.

Additional comments

It is not clear to me where penguins' colonies were located on Ardley Islands in the past. Today, they seem located on its eastern side, while Ardley Lake stands on its western side. The lake seems presently separated from the active colonies by "mountains". As there is no information on the draining basin of Ardley Lake it is difficult to know whether the location of active colonies can bias the ornithogenic records. I mean no signal in the lake despite the presence of an important colony because the ornithological soil are not transported in the lake.

We have added a new Figure 6 to show past Ardley Island colony locations and how they relate to changing relative sea levels through the Holocene.

Line 224: Authors argue that volcanic eruptions induced "high local mortality". I did not get whether they are macro-evidences (more bones) to support this. Can't we imagine that penguins just moved to another nesting site?

Figure 2 demonstrates two evident clusters in Phase 4. The more ancient cluster appears related to clusters found during CONISS zones 6 and 8 when low REE are encountered. Is Phase 4"a" really a high occupation period?

Phase 4a satisfies our definition of a guano-phase: 'Theoretically, all *Fo.sed.* values >0% provide evidence for guano input above the local bedrock-terrigenous background level, but we used a weighted mean *Fo.sed.* value of >10% at the cut-off for CONISS-defined geochemical zone to indicate the sustained presence of penguins around Ardley Lake (Table 1, Supplementary Table 7)' p.27

Figure 5 shows out-of-phase tephra in Ardley Lake and nearby Yanou Lake. Pretty weird.

Why? Dating uncertainties?

This aspect was largely resolved by new age-depth modelling in the revised version. See our response overview, Figures 4, 5, Supplementary Notes 7 and 9 and Supplementary Discussion for further details. To summarise, the alignment of the tephra ages between the ARD and YAN records and to published ages improved significantly when we included new penguin bone-collagen ages in the ARD-M5 age depth model.

Phase 3 is not represented by a white box in P. sp guano (Fos >10%).

This diagram combined species and locational information, which was confusing. We have constructed new diagrams for the main text, which more clearly show how key results from ARD and YAN records relate to each other and other records.

MCA marked during cool air temperatures and high sea ice presence. Weird.

The timing of MCA warming is from Bentley et al.⁹ which was a compilation of Antarctic Peninsula Holocene palaeoclimate data available at the time. Summarised timings for the MHH, MCA and RRR warmer periods are marked on this diagram. The Bentley et al.⁹ paper followed, and broadly agreed with, a similar exercise undertaken by Ingólfsson et al.¹⁸. In the Bentley et al.⁹ paper, we stated that evidence for the MCA on the Peninsula was not as strong as the MHH, and it was not present in all records. We retain the Bentley et al.⁹

overview to allow comparison with new Yanou Lake temperature data, but show the MCA as a dotted line in Figure 4.

Supplementary material

Chronology

It is noted that freshwater samples were calibrated with OXCAL why marine samples were calibrated using marine13. Was it in OXCAL too? It is then stated that the age model was developed with BACON. Were the calibrated ages injected into BACON? If so, why not using BACON to calibrate the raw ^{14}C ages? If not, why first using OXCAL?

We have clarified how radiocarbon dating, calibration and age-depth modelling were undertaken in the revised Methods and Supplementary Methods. Supplementary Note 7 describes how BACON age-depth modelling works in more detail. To summarise:

- The statistics for the data table were generated in OXCAL using raw (i.e., as-measured/uncalibrated) ^{14}C ages and their $1\text{-}\sigma$ measurement error and MARINE13 and SH13 curves.
- The age-depth models were generated in BACON inputting raw ^{14}C ages. These were calibrated within BACON using the MARINE13 and SH13 curves specified in the coding.
- OXCAL, like CALIB, can also be used as a radiocarbon calibration program for individual samples, and generates a useful summary set of statistics for each sample.
- BACON and OXCAL can be used as Bayesian age-depth modelling programs.
- Both BACON and OXCAL age-depth models are more statistically valid than simple linear regression or non-Bayesian and point-point age-depth modelling.
- We used the R-based BACON program for age-depth modelling

It is strange that the total reservoir age correction is 1870 years from the Anvers Shelf record (quite offshore location) and 1064 years in the marine part of the lake core (probably semi-enclosed basin). With old ice runoffs and slow ocean circulation south of South Shetland Islands I would have expected the opposite. Let's suggest that the AR correction is too small and was around 1900 years as in W-AAP core, what will it change to the interpretations vs climate?

We address this point in Supplementary Note 5 and in the Supplementary Methods (Anvers Trough Record section) in the revised version. To summarise, Figure 2, below, shows that surface sediment radiocarbon offsets around the SSI are in fact smaller than, generally speaking, those in cores taken further south and further offshore from the AP. Moreover, modern-day surface water radiocarbon ages from samples taken in Maxwell Bay on the SSI have a weighted mean measured radiocarbon age of $1,064 \pm 10$ years. This offset was used in Milliken et al. (2009)⁸ and was based on data in Table 1, below, supplied by Julia Wellner (pers. comm.) and used here with permission. The ΔR value of 700 ± 50 years used by Emslie et al.^{6,19} and in the revised version of this paper is therefore more appropriate for the South Shetland Islands as it more closely matches several modern day surface water ages from several sites in Maxwell Bay. See Supplementary Note 5 for further details. In the Yanou terrestrial record, single species moss ages return a near zero age in surface sediments and we consider these to be the most reliable ages. Dating the same species of moss in terrestrial sediments directly above the marine part of the core places an upper limit on the scale of the marine reservoir correction that can be applied to marine sediments in the Yanou Lake core. Applying a marine reservoir correction of 1,900 years would create unrealistic age reversals in the age-depth profile. For the Anvers core, we followed the standard marine coring practice of applying an age correction of 1,870 years, which corresponds to age of the surface sediments (15 in Figure 2 below). This offset is consistent with older ages from cores further offshore (16, below), and younger surface sediment ages from nearer-shore cores 4–7, 14 (see Figure below).

Table 1. Surface water radiocarbon ages for Maxwell Bay (supplied by Julia Wellner and used with permission).

Core water depth (m)	^{14}C age	Core water depth (m)	^{14}C age
---------------------	----------------------	---------------------

Maxwell Bay

Bransfield Strait

150	1,000 ± 25	80	1,130 ± 25
350	1,060 ± 20	251	1,070 ± 25
477	1,070 ± 30	464	1,060 ± 25

Figure 2. (Anna Hey Ph.D. Thesis²⁰): Map of WAP showing spatial variability in published surface sediment ages. Core numbering refers to Table A2.1, where details of publication, sample interval and conventional ¹⁴C age the can be found.

Sea ice record

Fcurta and Fkerguelensis records are generally anti-correlated in Southern Ocean records. What is here the point to present their ratio that may only increase the amplitude of the Holocene variations? I would like to see their raw data along with other species. Fcylindrus, Tantarctica1 and Tantarctica2 for example for a better assessment of local environmental conditions.

The Anvers Trough core data is a subset of the dataset and will be presented in a forthcoming article: Allen, C.S., Pike, J. Hey, A.R., Hodgson, D. A. and Crosta, X. (to be submitted)²¹. This paper develops themes outlined in this paper in more detail and includes marine diatom records of Holocene palaeoclimate along the West and North-East Antarctic Peninsula. Claire Allen is happy to supply the whole dataset for the Anvers Trough if required (csall@bas.ac.uk). In the revised version, we included a summary of the main diatom-based environmental groupings from the Anvers Trough core (Supplementary Fig. 15). The pelagic open water species plot included in Figure 4 corroborates the $F_{curta}/F_{kerguelensis}$ plot, supporting our original interpretation. We used the $F_{curta}/F_{kerguelensis}$ ratio as it relates to two sea-ice proximal species and is thought

to reflect the amount of UCDW on the continental shelf in the Anvers Trough core location. It represents the dominant control on diatom production associated with the open ocean (low values) versus productivity blooms associated with increased sea-ice cover at a particular site. The ratio has been used previously by Buffen et al.²², Massé et al.²³, Denis et al.²⁴, Kim et al.²⁵ and Peck et al.²⁶ and provides a useful amalgamation of three proxies for sea-ice extent (winter, spring, summer sea ice proxies) which largely reflect the length of sea-ice cover season. The point of including them as a ratio is to increase the amplitude of the signal and summarise the information contained.

Other

Phase 4 is noted between 111.9–67.6 cm in Supp. Table 5 while between 155–67.6 cm in Figure 2. This confirms my previous comment that the Gentoo colony was probably not that large during Phase 4 "a".

The shading used in the original Table S5 was not correct. Zone 9 (Phase 4a) should also have been shaded a similar colour as the rest of Guano Zone 4. We have revised and corrected this table, taking into account age changes associated with the new ARD-M5 age-depth model used. We also now include summary guano-phase statistics that clearly show guano phase 4a meets our definition of a guano phase since its weighted mean $F_{o, sed}$ value is 12.95 (i.e., >10%) (Table 1 and Supplementary Table 7). We set the lower threshold value at 10% to account for measurement errors and to exclude the possibility of guano deposition by a small number of passing penguins or skuas nesting in the catchment.

References:

1. Wang, J., Wang, Y., Wang, X. & Sun, L. Penguins and vegetations on Ardley Island, Antarctica: evolution in the past 2,400 years. *Polar Biol.* **30**, 1475–1481 (2007).
2. Sun, L., Xie, Z. & Zhao, J. A 3,000-year record of penguin populations. *Nature* **407**, 858 (2000).
3. Sun, L. G., Xie, Z. Q. & Zhao, J. The sediments of lake on the Ardley Island, Antarctica: identification of penguin-dropping soil. *Chinese J. Polar Sci.* **12**, 1–8 (2001).
4. Clapperton, C. M., Sugden, D. E., Birnie, J. & Wilson, M. J. Late-glacial and Holocene glacier fluctuations and environmental change on South Georgia, Southern Ocean. *Quat. Res.* **31**, 210–228 (1989).
5. Del Valle, R. A., Montalti, D. & Inbar, M. Mid-Holocene macrofossil-bearing raised marine beaches at Potter Peninsula, King George Island, South Shetland Islands. *Antarct. Sci.* **14**, 263–269 (2002).
6. Emslie, S. D., Polito, M. J. & Patterson, W. P. Stable isotope analysis of ancient and modern gentoo penguin egg membrane and the krill surplus hypothesis in Antarctica. *Antarct. Sci.* **25**, 213–218 (2013).
7. Casanovas, P., Lynch, H. J. & Fagan, W. F. Multi-scale patterns of moss and lichen richness on the Antarctic Peninsula. *Ecography* **36**, 209–219 (2013).
8. Milliken, K. T., Anderson, J. B., Wellner, J. S., Bohaty, S. M. & Manley, P. L. High-resolution Holocene climate record from Maxwell Bay, South Shetland Islands, Antarctica. *Geol. Soc. Am. Bull.* **121**, 1711–1725 (2009).
9. Bentley, M. J. *et al.* Mechanisms of Holocene palaeoenvironmental change in the Antarctic Peninsula region. *The Holocene* **19**, 51–69 (2009).
10. Sterken, M. *et al.* Holocene glacial and climate history of Prince Gustav Channel, northeastern Antarctic Peninsula. *Quat. Sci. Rev.* **31**, 93–111 (2012).
11. Bentley, M. J. *et al.* A community-based geological reconstruction of Antarctic Ice Sheet deglaciation since the Last Glacial Maximum. *Quat. Sci. Rev.* **100**, 1–9 (2014).
12. Ó Cofaigh, C. *et al.* Reconstruction of ice-sheet changes in the Antarctic Peninsula since the Last Glacial Maximum. *Quat. Sci. Rev.* **100**, 87–110 (2014).
13. Etourneau, J. *et al.* Holocene climate variations in the western Antarctic Peninsula: evidence for sea ice extent predominantly controlled by changes in insolation and ENSO variability. *Clim. Past* **9**, 1431–1446 (2013).
14. Yoon, H. II, Park, B.-K. K., Kim, Y. & Kim, D. Glaciomarine sedimentation and its paleoceanographic implications along the fjord margins in the South Shetland Islands, Antarctica during the last 6000 years. *Palaeogeogr. Palaeoclimatol. Palaeoecol.* **157**, 189–211 (2000).
15. Hass, H. C., Kuhn, G., Monien, P., Brumsack, H.-J. & Forwick, M. Climate fluctuations during the past two millennia as recorded in sediments from Maxwell Bay, South Shetland Islands, West Antarctica. *Geol. Soc. London, Spec. Publ.* **344**, 243–260 (2010).

16. Monien, P., Schnetger, B., Brumsack, H.-J., Hass, H. C. & Kuhn, G. A geochemical record of late Holocene palaeoenvironmental changes at King George Island (maritime Antarctica). *Antarct. Sci.* **23**, 255–267 (2011).
17. Barbara, L. *et al.* Environmental responses of the Northeast Antarctic Peninsula to the Holocene climate variability. *Paleoceanography* **31**, 131–147 (2016).
18. Ingólfsson, Ó. & Hjort, C. Glacial history of the Antarctic Peninsula since the Last Glacial maximum - a synthesis. *Polar Res.* **21**, 227–234 (2002).
19. Emslie, S. D. Radiocarbon dates from abandoned penguin colonies in the Antarctic Peninsula region. *Antarct. Sci.* **13**, 289–295 (2001).
20. Hey, A. R. Palaeoclimate reconstructions from the Antarctic Peninsula; linking marine and terrestrial records. (Cardiff University, 2009).
21. Allen, C. S., Pike, J., Hey, A. R., Hodgson, D. A. & Crosta, X. Marine diatom records of Holocene palaeoclimate along the West and North-East Antarctic Peninsula.
22. Buffen, A., Leventer, A., Rubin, A. & Hutchins, T. Diatom assemblages in surface sediments of the northwestern Weddell Sea, Antarctic Peninsula. *Mar. Micropaleontol.* **62**, 7–30 (2007).
23. Massé, G. *et al.* Highly branched isoprenoids as proxies for variable sea ice conditions in the Southern Ocean. *Antarct. Sci.* **23**, 487–498 (2011).
24. Denis, D. *et al.* Holocene glacier and deep water dynamics, Adélie Land region, East Antarctica. *Quat. Sci. Rev.* **28**, 1291–1303 (2009).
25. Kim, J.-H. *et al.* Holocene subsurface temperature variability in the eastern Antarctic continental margin. *Geophys. Res. Lett.* **39**, n/a–n/a (2012).
26. Peck, V. L., Allen, C. S., Kender, S., McClymont, E. L. & Hodgson, D. A. Oceanographic variability on the West Antarctic Peninsula during the Holocene and the influence of upper circumpolar deep water. *Quat. Sci. Rev.* **119**, 54–65 (2015).

REVIEWERS' COMMENTS:

Reviewer #1 (Remarks to the Author):

The authors have done a very thorough job in responding to my review on this paper. I agree with all the changes and they have adequately addressed all questions and concerns. I believe the paper is much improved and worthy of publication. It will be a valuable contribution to the literature on penguin occupation history in the Antarctic Peninsula.

Reviewer #3 (Remarks to the Author):

Monien and co-authors (Roberts et al. now) here present a greatly improved revised version that thoroughly answers most of my comments. I reckon that they have done an impressive amount of work to better deconvolute the impacts of climate and volcanic activity on penguin population dynamics over the Holocene in northwestern Antarctic Peninsula. The new data (GDGT; refined tephra and grain size data) and the statistical analyses (Supp. Material) are essential and very supportive of their hypothesis. I am now happy how the authors used both climate and volcanic eruptions to explain changes in Holocene penguin population even though we can endlessly argue about which climate records are more representative of global vs regional changes. I am also happy that more information is given on the relationships between volcanic eruptions and colony dynamics (intensity – VEI; number of eruptions – tephra layers).

The paper is very well structured, written and illustrated. I am sure that it will be of interest to a very large audience. I advise publication after minor revision following the points mentioned below.

Line 163 and Fig 1. Summer sea ice limit cannot lie over the middle part of Bransfield Strait. It would mean that the region south of the summer sea ice limit is sea ice covered year round. This is wrong as shown in many papers (Cavalieri and Parkinson, 2008; Stammerjohn et al., 2008; Comiso et al., 2011; Hobbs et al., 2016) and satellite images (<http://www.seaice.dk/latest/todays-amsr2-s-mos.jpg>). Summer sea ice only persists south of the Antarctic Peninsula (cf image at the link above).

2016 November 23 image of sea ice cover in Antarctica => already no sea ice in Bransfield Strait at mid-spring.

Guano phases result from the presence of large penguin populations on Ardley Island. I guess that high populations occur when nesting (warmer surface air temperatures) and feeding (warmer sea-surface temperatures and lower sea-ice cover) co-exist. Within the climate part, authors can probably make a better use of paleo-climate and paleo-oceanic data to interpret past population dynamics in terms of nesting and/or foraging success.

Fig 1a. Put north upward (90° clockwise rotation) that all maps are in the same direction.

We thank both reviewers for taking the time to provide constructive and helpful comments on the initial and revised versions of this paper. We respond to specific comments from Reviewer 3 in bold below.

REVIEWERS' COMMENTS:

Reviewer #1 (Remarks to the Author):

The authors have done a very thorough job in responding to my review on this paper. I agree with all the changes and they have adequately addressed all questions and concerns. I believe the paper is much improved and worthy of publication. It will be a valuable contribution to the literature on penguin occupation history in the Antarctic Peninsula.

Reviewer #3 (Remarks to the Author):

Monien and co-authors (Roberts et al. now) here present a greatly improved revised version that thoroughly answers most of my comments. I reckon that they have done an impressive amount of work to better deconvolute the impacts of climate and volcanic activity on penguin population dynamics over the Holocene in northwestern Antarctic Peninsula. The new data (GDGT; refined tephra and grain size data) and the statistical analyses (Supp. Material) are essential and very supportive of their hypothesis. I am now happy how the authors used both climate and volcanic eruptions to explain changes in Holocene penguin population even though we can endlessly argue about which climate records are more representative of global vs regional changes. I am also happy that more information is given on the relationships between volcanic eruptions and colony dynamics (intensity – VEI; number of eruptions – tephra layers).

The paper is very well structured, written and illustrated. I am sure that it will be of interest to a very large audience. I advise publication after minor revision following the points mentioned below.

Line 163 and Fig 1. Summer sea ice limit cannot lie over the middle part of Bransfield Strait. It would mean that the region south of the summer sea ice limit is sea ice covered year round. This is wrong as shown in many papers (Cavaliere and Parkinson, 2008; Stammerjohn et al., 2008; Comiso et al., 2011; Hobbs et al., 2016) and satellite images (<http://www.seaice.dk/latest/todays-amsr2-s-mos.jpg>). Summer sea ice only persists south of the Antarctic Peninsula (cf image at the link above). 2016 November 23 image of sea ice cover in Antarctica => already no sea ice in Bransfield Strait at mid-spring.

The corresponding authors agree with Reviewer 3 on this point and have updated Fig. 1a as suggested. The sea ice data plotted in the revised version of Fig. 1a was produced by MAGIC (Mapping And Geographic Information Centre) at BAS using sea-ice duration data calculated from daily remote sensed readings that were averaged into monthly mean values and then into a single grid for the period 1979–2007. However, in effect, this process took the mean of Oct–Feb limit data and produced a single summer limit for this part of Antarctica running all the way up through the centre of the Bransfield Strait. As Reviewer 3 correctly identifies, and as our new Fig 1a shows, this approach does not reflect any of the median spring-summer sea-ice limits between Sept–Feb. Hence, we now show the National Snow and Ice Data Center (NSIDC) 1981–2010 median maximum monthly sea-ice positions, with the winter-peak conditions (Sept.) through to the

median minimum summer (Feb.) extent (for 20% sea-ice coverage) shown as solid lines. This more accurately illustrates how our field sites on King George Island become sea-ice free before other parts of the Antarctic Peninsula, enabling nearshore shelf-edge foraging earlier in the season around the South Shetland Islands. It also highlights that our Anvers trough record is (on average) free of sea-ice cover about a month earlier than the Palmer Deep area, possibly explaining why our new sea-ice record does not always correspond with sea-ice and temperature records from inner-shelf areas on the western and northeastern Peninsula. Data were downloaded and referenced as from https://nsidc.org/data/seaice_index/.

Guano phases result from the presence of large penguin populations on Ardley Island. I guess that high populations occur when nesting (warmer surface air temperatures) and feeding (warmer sea-surface temperatures and lower sea-ice cover) co-exist. Within the climate part, authors can probably make a better use of paleo-climate and paleo-oceanic data to interpret past population dynamics in terms of nesting and/or foraging success.

Some parts of a more detailed paragraph covering the relationship between changing environmental conditions and its impact on nesting/foraging success were moved to Supplementary Information when we refocused the paper around the volcanic aspect of the study (as requested by all reviewers). As suggested above, we have updated our summary conclusions paragraph to reflect the fact that, once deglaciated, climatic conditions on Ardley Island throughout the Holocene were generally amenable enough for sustained (gentoo) penguin habitation. In general, the Ardley Lake colony population maxima relate to enhanced long-term foraging and nesting success, determined by a combination of relatively elevated and sustained above 6-0 ka average temperatures, with comparatively storm-free conditions (leading to reduced terrestrial snow and ice cover), and around or just below 6-0 ka average sea-ice cover/seasonal extent. However, the last three mid-late Holocene guano-phases were interrupted by volcanic activity from Deception Island, which was at least an order of magnitude greater than present-day eruptions and deposited a layer of volcanic ash across the landscape that disrupted nesting/foraging activities for several years (possibly several decades). Therefore, across the Holocene as a whole, we find no consistent relationships with local-regional atmospheric and ocean temperatures or sea-ice conditions.

At this stage, we feel that it is probably too speculative to expand further beyond this conclusion. This is mainly because the environmental and nesting/foraging relationships in the Ardley Lake record likely relate to gentoo colonies, which were sporadically disrupted by volcanic activity. Therefore, Ardley Island might not necessarily be representative of wider Peninsula or cross-species penguin behaviour. We are in the process of obtaining more long-term geochemical, biomarker and genetic evidence from other sites across the Peninsula region to further test links between past changes in climate, sea-ice and foraging/nesting, including, if feasible, how these relationships could vary between different species in the past.

Fig 1a. Put north upward (90° clockwise rotation) that all maps are in the same direction.

The map originally used in Fig 1a was, by default in ARC, a South Pole Stereographic projection. Figs. 1b-d are projected in WGS_1984_UTM_Zone21S, Transverse Mercator. As suggested, all maps in Fig. 1 now have the same projection and orientation.

We have also made all formatting changes requested as outlined in the returned marked up PDF (as replies to comments) and made the following minor corrections/improvements:

- 1) Added a visual summary and description of bones found in the Ardley Lake core to Figure 2 from Supplementary Figure 4 to bring this aspect of the study more to the forefront in the main body of the paper
- 2) The font size in Figure 1 is 12 pt for main/key labels and 9 pt for all other text. Font sizes in other diagrams are similar (10pt where possible/usually, but we have also used 8 pt only when there was insufficient room). We have also made font sizes and consistent across main diagrams.
- 3) Simplified and 'decluttered' Figures 4 and 5 as requested
- 4) Added a RSL summary diagram to Figure 6 to make this a standalone figure that doesn't need referring back to Figure 5
- 5) Rearranged Figure 6 into a grid format as requested
- 6) We have made Word versions of Tables 1 and 2 - these were originally constructed in Excel – we include PDF / original Excel versions showing how these tables were intended to look in case they are need to be reformatting
- 7) The abstract is 164 words, at the 164 maximum word limit
- 8) The main text is 4813 words, within the 5000 word limit
- 9) We have retained the same number of references (in the same positions as before) in the main text
- 10) Because of the large number of diagrams in the Supplementary Information, we were not able to produce a Word document with a reasonable file size. As requested, instead, we have produced a single PDF, which is 32 Mb. We can supply a word version of the SI text if it needs editing.